

# A WATERSHED CLASSIFICATION APPROACH THAT LOOKS BEYOND HYDROLOGY: APPLICATION TO A SEMI-ARID, AGRICULTURAL REGION IN CANADA

Jared D. Wolfe[1*], Kevin R. Shook[2], Chris Spence[3], Colin J. Whitfield[1,4]

[1]Global Institute for Water Security, University of Saskatchewan, Saskatoon, Saskatchewan, Canada
[2]Centre for Hydrology, Saskatoon, Saskatchewan, Canada
[3]National Hydrology Research Centre, Environment and Climate Change Canada, Saskatoon, Saskatchewan, Canada
[4]School of Environment and Sustainability, University of Saskatchewan, Saskatoon, Saskatchewan, Canada

*corresponding author: jared.wolfe@usask.ca

**ABSTRACT**

17          Classification and clustering approaches provide a means to group watersheds according
to similar attributes, functions, or behaviours, and can aid in managing natural resources.
Although they are widely used, approaches based on hydrological response parameters restrict
analyses to regions where well-developed hydrological records exist, and overlook factors
contributing to other management concerns, including biogeochemistry and ecology. In the
Canadian Prairie, hydrometric gauging is sparse and often seasonal. Moreover, large areas are
endorheic and the landscape is highly modified by human activity, complicating classification
based solely on hydrological parameters. We compiled climate, geological, topographical, and
land cover data from the Prairie and conducted a classification of watersheds using a hierarchical
clustering of principal components. Seven classes were identified based on the clustering of
watersheds, including those distinguishing southern Manitoba, the pothole region, river valleys,
and grasslands. Important defining variables were climate, elevation, surficial geology, wetland
distribution, and land cover. In particular, three classes occur almost exclusively within regions
that tend not to contribute to major river systems, and collectively encompass the majority of the
study area. The gross difference in key characteristics across the classes suggests that future
water management and climate change may carry with them heterogeneous sets of implications
for water security across the Prairie. This emphasizes the importance of developing management
strategies that target sub-regions expected to behave coherently as current human-induced
changes to the landscape will affect how watersheds react to change. The study provides the first
classification of watersheds within the Prairie based on climatic and biophysical attributes, with
the framework used being applicable to other regions where hydrometric data are sparse. Our
findings provide a foundation for addressing questions related to hydrological, biogeochemical,
and ecological behaviours at a regional level, enhancing the capacity to address issues of water
security.

**A WATERSHED CLASSIFICATION APPROACH THAT LOOKS BEYOND HYDROLOGY: APPLICATION TO A SEMI-ARID, AGRICULTURAL REGION IN CANADA**

**1. INTRODUCTION**

Watershed classification methods provide a means of grouping watersheds according to similar attributes, or behaviours, and can identify sub-regions that are expected to exhibit coherent responses. This strategy can identify how catchment characteristics are similar, or dissimilar, among groups of watersheds and thus might influence hydrologic behaviour (McDonnell and Woods, 2004). Classifying watersheds can be useful for developing predictions in ungauged basins (Peters et al., 2012), and moreover, classification can be used to inform how changes to key traits (e.g., climate and land management) may affect system function. Establishing these links between watershed function and biophysical structure, including hydroclimate, is an opportunity of watershed classification (Wagener et al., 2007). Accordingly, the regionalization of hydrological response through watershed classifications has been used to inform natural resource management (Detenbeck et al., 2000; Jones et al., 2014).

Many different approaches to watershed classification have been employed to date, including non-linear dimension reduction techniques (Kanishka and Eldho, 2017), decision trees (Bulley et al. 2008), and independent component analysis (Mwale et al., 2011), among others. Hydrological characteristics (e.g., statistical properties of streamflow regime) are widely used to inform classification owing to their potential linkages between watershed features and hydrologic responses (Brown et al., 2014; Sivakumar et al., 2013; Spence and Saso, 2005). Other classification exercises have included a wider number of characteristics, including biophysical attributes along with streamflow response, to differentiate watershed classes (e.g., Sawicz et al., 2014; Burn, 1990). Ecoregions, which incorporate historical aspects of climate, topography, and vegetation regimes, have also served as a method of differentiation for eco-hydrological studies (Loveland and Merchant, 2004). In select cases, classification is performed independently of streamflow response factors (Knoben et al., 2018). In arid or poorly gauged regions of the world, these types of approaches to classification that are independent from or not strongly dependent on hydrological indices (streamflow response), are needed, although few such classifications

have been performed. The need for new approaches to watershed classification can also be true
of regions undergoing strong pressures from climate change and land-use, where historical
streamflow records may not reflect current behaviour, particularly if a regime shift has occurred.
In Canada, watershed classification has been applied in many regions (e.g., Cavadias et
al., 2001; Ouarda et al., 2002; Spence and Saso, 2005). To date, most have focused on larger
basins, and none have covered in detail the semi-arid Canadian Prairie, which spans nearly 5 x
$10^5$ km$^2$ in western Canada, from the Rocky Mountain foothills in the west to Lake Winnipeg in
the east (Fig. 1). This is despite its importance as a major food producing region of the world and
one that faces numerous water security challenges (Gober and Wheater, 2014; Spence et al.,
2018). Earlier work by Durrant and Blackwell (1959) grouped large Prairie watersheds based on
flood regimes. A recent classification that included the Prairie region focused on stream
hydrology (e.g., MacCulloch and Whitfield, 2012) but was broader and included watersheds
from mountainous and forested regions to the west and north, respectively. In the Canadian
Prairie, and similar regions elsewhere, extrapolating catchment-scale field and modelling studies
presents challenges. It is inherently difficult to explain or predict different responses among
basins, as poorly developed stream networks with intermittent or seasonal flow do not easily lend
themselves to classification methods featuring streamflow response. MacCulloch and Whitfield
(2012), who found a single streamflow class across the Canadian Prairie, raised the question as
to whether a single grouping is appropriate, and suggested the need to expand classifications to
include a greater diversity of biological, physical and chemical properties.
Like many of the world's agricultural regions, the Canadian Prairie has undergone vast
environmental change co-incident with the green revolution. Predominant agricultural practices
have changed over the decades, and each is known to influence water cycling and storage,
including tillage practices, summer fallowing, and cropping type (Awada et al., 2014; Van der
Kamp et al., 2003; Shook et al., 2015). Significant warming over the last 70 years, especially in
winter (Coles et al., 2017; DeBeer et al., 2016) has resulted in more rain at the expense of snow
(Vincent et al., 2015), and multiple-day rainfall events have been increasing in frequency relative
to shorter events in some regions (Dumanski et al., 2015; Shook and Pomeroy, 2012). These
observed changes in precipitation have reduced the predictability of runoff derived from
snowmelt, and add uncertainty to water management and agricultural decision-making.
Disentangling the relative impacts of climate and land-use changes on water quantity and
quality is complex, particularly as their effects are heterogeneous across spatial extent and scale.
For the Prairie and elsewhere, new approaches to classification that can distinguish sub-regional
and, importantly, sub-hydrometric station variability, are needed. Further, because land
management decisions in agricultural regions are intrinsically linked to system function, there is
a need for classifications that can inform decision-makers at a relevant scale. Indeed, stable
isotope-based investigations of runoff from small lake catchments in the Boreal Plains (north of
the Prairie) emphasize the need for local-scale characterization of watershed behaviour (Gibson
et al., 2010, 2016), while streamflow dynamics for the Prairie and nearby Boreal Plain are linked
to local surface geology and land cover (Devito et al., 2005; Mwale et al., 2011), suggesting an
opportunity for a new approach to watershed classification in the region. Another potential
advantage of a more comprehensive approach is that by de-emphasizing available hydrometric
observations for larger and well-studied or monitored basins and including other environmental
characteristics, the risk of overlooking other functions (e.g., ecology, biogeochemistry) that may
be equally important to the management of a watershed's natural resources can be reduced. A
system-based watershed classification for the Prairie that avoids the prejudice of classifying only
those watersheds where a reasonably robust understanding of hydrology or streamflow exists can
serve as a template for other regions of the world where streamflow-based classification is not
viable.
The objective of the present work is to develop a watershed classification system based
on hydrologically and ecologically significant traits for the Canadian Prairie. In this region,
assessment of localized hydrological response to change is challenged by limited spatial
resolution of observed streamflow data, and higher order streamflow being unrepresentative of
local response due to a poorly-developed drainage network. In establishing such an approach, we
seek to advance our understanding of watershed hydrology and broader watershed behaviour
within the Prairie whilst also providing a framework for similar classification exercises in other
regions where streamflow-based methods are not ideal. Our approach avoids the limitations of
classifying according to known hydrologic response, and increases the spatial resolution of
watershed classification relative to many existing approaches. We compile physiographic
characteristics, including geology, wetland distribution, and land cover, of watersheds
approximately 100 km$^2$ to achieve the classification. This framework will identify those areas
that are climatically and geographically similar, and thus might be expected to respond in a
hydrologically coherent manner to climate and land management changes. Additionally, it
provides a foundation on which to base prediction of watershed hydrologic, biogeochemical and
ecological responses to these stressors.

**2. DATA COLLECTION & COMPILATION**

*2.1. Region domain and description*

The Canadian Prairie (Prairies ecozone) spans the provinces of Alberta, Saskatchewan,

and Manitoba, and is part of the Nelson Drainage Basin (Fig. 1). Climate is semi-arid, with mean
annual precipitation ranging between 350 and 610 mm (1970–2000) increasing from west to east.
Mean annual temperature was 1–6°C over the same period with warmer conditions towards the
southwest (Mekis and Vincent, 2011; Vincent et al., 2012;
http://climate.weather.gc.ca/climate_normals/index_e.html). Much of the region deglaciated
during the Late Pleistocene approximately 10,000 years before present, resulting in an often
hummocky landscape with numerous depressions. Combined with the dry climate, the relatively
short post-glaciation history has prevented maturing of a ubiquitous drainage network, and many
headwaters remain disconnected from higher order streams (Shook et al., 2015). Depressions in
the hummocky landscape, and the wetlands that form within them, are important features for
Prairie hydrology (Van der Kamp et al., 2016) and often facilitate groundwater recharge (i.e.,
depression-focused recharge) (Van der Kamp and Hayashi, 2009). The location of wetlands and
their size, relative to the watershed outlet controls hydrologic gate-keeping (e.g., Spence and
Woo, 2003), and thus the potential to contribute streamflow to higher-order watersheds
(Leibowitz et al., 2016; Shaw et al., 2012; Shook et al., 2013). The size distribution of wetlands
within a watershed and their spatial arrangement also dictate biogeochemical function and
provide habitat and foraging for biota (Evenson et al., 2018). Terrestrial vegetation is typically
open grassland, with aspen parkland ecotone along the northern edges of the ecozone boundary
(Ecological Stratification Working Group, 1995).

*2.2. Watershed boundaries*
The focus of this study was on those watersheds that drain a distinctively prairie
landscape, with watersheds defined according to topographic delineation. Thus, we constrained
our study to the Canadian Prairie ecozone ($4.7 \times 10^5$ km$^2$) and watersheds occurring therein.
Delineations of candidate study watersheds were obtained from the HydroSHEDS global dataset
(Lehner and Grill, 2013). Watershed boundaries within this dataset were based on Shuttle Radar
Topographic Mission (SRTM) digital elevation model (DEM) calculated at a 15 arc-second
resolution. The resolution is equivalent to for example approximately 285 m east-west and 464 m
north-south at Saskatoon, SK. As with other SRTM products, the HydroSHEDs dataset may be
prone to errors in regions with low relief due to elevation precision of 1 m. However, the dataset
provided watershed delineations over the geographic region of interest and at a fine enough scale
(i.e., 100 km$^2$), and thus, it was sufficient based on data availability for purpose of the current
study.

Only those watersheds completely within the Canadian Prairie ecozone were extracted (n

= 4729) from the HydroSHEDs dataset. Those watersheds that were very large (>4000 km$^2$) or
small (<5 km$^2$) were removed from analysis (see Table S1). Because HydroSHEDs includes the
basins of larger water bodies, including lakes, watersheds consisting of a majority of water were
removed as the study only concerns the uplands of these systems. Finally, highly urbanized areas
(i.e., watersheds with cover being >40% urban) were removed. After considering these criteria,
4175 watersheds remained for use in subsequent analyses, covering a total area of $4.2 \times 10^5$ km$^2$.
Mean watershed area for this subset was $99.8 \pm 58.7$ km$^2$.

*2.3. Physiographic data collection*

The physiographic watershed variables were assembled from Canadian provincial and

federal governments and non-governmental agency datasets (see Table S2 for a full list of
variables and their sources). Variables were derived from climatic, hydrologic, geological,
geographic, and land cover data, and details are described briefly below. Spatial processing and
statistical analyses were conducted in ArcGIS version 10.5 and R version 3.4.3 (R Core Team,
2018), respectively.

*2.3.1. Climate*
Mean annual precipitation and temperature data were derived from the Canadian Gridded
Temperature and Precipitation Anomalies (CANGRD) dataset spanning (ECCC, 2017).
CANGRD is the only gridded climate product available for the region that uses adjusted and
homogenized station data, and was picked for this reason (Mekis and Vincent, 2011; Vincent et
al., 2012). The 1970–2000 period was chosen because the number of stations with adjusted and
homogenized data used to derive CANGRD significantly diminished after 2000 (Laudon et al.,
2017). Mean annual values over the 30-year period were constructed from 50 km resolution
gridded cells (n = 626) within and surrounding the Prairie ecozone, and interpolated to a higher
spatial resolution raster by kriging using a spherical semivariogram. Values were clipped
according to the watershed boundaries, and averaged over the watersheds to obtain mean annual
precipitation and temperature for each watershed. Mean annual potential evapotranspiration
(PET) was derived as a measure of dryness across the region. To maintain consistency among
climate data, and use the same temperature data as described above, options were limited with
which to calculate PET. The Thornthwaite equation (Thornthwaite, 1948) was applied using the
R package *SPEI* (Vicente-Serrano et al., 2010). A disadvantage of the Thornthwaite approach is
that it calculates PET solely as a function of air temperature and latitudinal position, and it
assumes a fixed correlation between temperature and radiative forcing. As such, it integrates
effects of other factors directly or indirectly influencing radiation or latent heat, like advection,
vegetation, and humidity. The calculation adjusts for any lag in this relationship using
corrections for latitude and month; however, it likely does not represent the full annual and
seasonal variability in PET across a landscape, given regional heterogeneity of the
aforementioned factors. Despite the limitations, the simplicity of this method is ideal for
application across the wide geographic area of interest with limited data required as input,
allowing for approximation of mean annual PET for the study area.

*2.3.2. Wetland traits*
Large regions within the Canadian Prairie have been designated as being "non-effective",
where they do not contribute flow to the stream network, at least one year in two (Godwin and
Martin, 1975). The location of these regions are shown in Figure 1. This definition stems from
work by Agriculture and Agri-Food Canada where prairie drainage areas were divided into *gross*
and *effective* drainage areas, whereby the former describes the area within a topographic divide
that is expected to contribute under highly wet conditions, and the latter is the area that
contributes runoff during a mean annual runoff event (Mowchenko and Meid, 1983). Thus, at its
simplest, the non-effective area is the difference between the gross and effective drainage area;
however, the exact area contributing runoff is dynamic and the controls complex, which include
antecedent storage capacity and climatic conditions (Shaw et al., 2012: Shook and Pomeroy,
2015). Briefly, the "non-effective" regions are caused by the intermittent connectivity of runoff
among the landscape depressions, which trap runoff, and prevent it from contributing to
downstream flow when the depressions are not connected. Trapped surface water can form
wetlands (hereafter, inclusively referring to water area ponded in these depressions). These
depressions can store water, and are indicative of water storage of the basin. Thus the non-
effective portion of a basin is an index of its lack of contribution and is an important quality
when considering the hydrological dynamics of this region (Shook et al., 2012).

The Global Surface Water dataset (Pekel et al., 2016) provides a geographically

comprehensive layer of any ~30 m x 30 m pixel that was inundated at least once between 1984
and 2015, as identified from the Landsat constellation of satellites. It was assumed that the
dataset was indicative of potential maximum wetland coverage, as this period spanned several
wet climate periods. As such, "wetland" in this context can include some seasonal ponds (i.e.,
prairie potholes) as well as larger or more permanent shallow water bodies (but see Section 2.2
and Table S1). Using the R package *raster* (Hijmans, 2017), wetland variables were calculated
for each study watershed, including fractional wetland area, and the number of wetlands within
the watershed per unit area (i.e., wetland density ($km^{-2}$)). The ratio of the area of the largest
wetland to total wetland area in the watershed was also used as a metric (i.e., $W_L$). Further, we
used the ratio of the linear distance of the largest wetland's centroid to the watershed outlet ($L_W$),
to the maximum watershed boundary distance to the outlet ($L_O$) to represent a centroid fraction
($L_W/L_O$; i.e., the relative location of the largest wetland to watershed outlet). The basin outlet was
defined as the point of lowest elevation on the watershed boundary. Both $W_L$ and $L_W/L_O$ can be
used to evaluate the relative importance of hydrological gate-keeping; for example, larger
wetland depressions located closer to the outlet control the likelihood of the watershed
contributing flow downstream and attenuating peakflow (Shook and Pomeroy, 2011; Ameli and
Creed, 2019).
To estimate wetland size distribution, it was assumed that they followed a Generalized
Pareto Distribution (GPD) defined according to (Shook et al., 2013):

$$F(z) = GPD(\mu, \beta, \xi) = 1 - \left[1 + \xi \left(\frac{z - \mu}{\beta}\right)^{-1/\xi}\right] \tag{1}$$


Where $z$ is wetland area, and $\mu$ is the location parameter (i.e., the minimum size for which the
distribution was fitted and has units of m$^2$), and the scale ($\beta$) and shape ($\xi$) parameters are
determined for each watershed. The $\beta$ parameter is an index of the dispersion of the distribution,
similar to the standard deviation, with the same units as the data being fitted (in this case m$^2$).
The $\xi$ parameter is dimensionless and governs the shape of the fitted distribution. Hosking and
Wallace (1987) plot the effect of variation in the shape parameter on the GPD. The scale and
shape parameters were used to quantify the size distribution of wetlands and thus to describe the
wetland frequency distributions for the cluster analyses (see 3.2). Note that because the sizes of
the water bodies were taken from infrequent remote-sensing measurements (i.e., the Landsat data
have a minimum revisit time of 8 or 16 days), they also are biased against short-lived water
bodies.

*2.2.3. Topographical parameters*
Topographic variables including the mean elevation, mean and coefficient of variation of
slope, and stream density were also calculated for each watershed. Because of the hummocky
nature of many regions in the domain, it is possible for a basin to have some fraction of its area
located at an elevation below that of the outlet. As such, the fraction of area below the basin
outlet ($A_{BO}$) was calculated for each basin. The elevation and slope variables were based on a
DEM generated from the SRTM dataset. Stream vectors were obtained from the Hydrographic
features CanVec (1:50000) series available from Natural Resources Canada
(https://open.canada.ca/data/en/dataset?q=canvec&sort=&collection=fgp). The total length of
streams within a watershed was calculated, and divided by the watershed area to produce the
stream density. Additionally, the dimension shape factor (DSF) was used to describe watershed
shape, as it has been found important for hydrological responses in previous Canadian catchment
classification exercises (Spence and Saso, 2005). The DSF (km$^{-1}$) was calculated as follows:

$$DSF = \frac{(0.28 \cdot P)}{A} \qquad (2)$$


Where $P$ (km) and $A$ (km$^2$) are the watershed perimeter and area, respectively, and derived from
the HydroSHEDS global dataset (Lehner and Grill, 2013).

Geographical parameters of surficial geology, local surface landforms, soil particle size

classes (sand, silt, clay), and soil zone were included in the analysis. Surficial geology polygons
were derived by compiling provincial government data sources for Alberta (Atkinson, 2017),
Saskatchewan (Simpson, 2008), and Manitoba (Matile, 2006). Due to the different geological
classification schemes for each province, more detailed classes were grouped to broader
categories related to depositional environment and surficial materials using those from the
Geological Survey of Canada (2014), which provided for comparison across provincial
boundaries. Local surface form (i.e., areas categorized by slope, relief, and morphology) and soil
zone data were obtained from the Soil Landscape dataset (AAFC, 2013). The soil zones in the
Canadian Prairie, used in the analyses were black, dark brown, brown, gray, and dark gray. The
zones incorporate characteristics of colour and organic content, which are influenced by regional
climate and vegetation. Clay, silt, and sand content were collected from the Detailed Soil Survey
of Canada (AAFC, 2015). Mean catchment values of surficial geology, local surface landform,
soil zone, and particle size class were determined by areal weighting of soil polygons within the
watershed boundaries.

*2.3.4. Land cover and cropland practice*

Fractional areas of land-use types were derived from the Agriculture and Agri-Food

Canada's 2016 Annual Crop Inventory (AAFC, 2016). These raster data define land-use and land
cover. Variables used in our analysis were standardized to watershed area and included
unmanaged grasslands, forests (i.e., the sum of coniferous, deciduous, and mixed forest areas),
pasture, and cropland (sum of cropped land areas). Predominant cropland practice was defined
according to the fractional area of tillage by agricultural region sub-division (e.g., normalized to
the area prepared for seed within that division by year). Averaged areas over the years 2011 and
2016 for each practice, including zero-till, conservation till (leaving crop residue on soil surface),
and conventional till (incorporating residues into soil) (Statistics Canada, 2016), were used to
describe these activities, and normalized as a fraction of the watershed.

*2.3.5. Hydrological variable calculation*

The relatively sparse hydrometric stream gauging in the domain, and the resulting paucity

of data, presents two notable challenges to hydrologic response-based watershed classification.
The first is that the basin network is biased to stations on higher-order (and often exotic) streams
traversing the region (i.e., larger river basins), and thus there are a limited number of
hydrometric gauges on streams draining solely Prairie watersheds, particularly at the spatial
resolution of our study watersheds (~100 km$^2$). Further, only a subset of these are considered
reference stations (i.e., gauging unmanaged flows). Second, in the more arid and/or cold regions
of the Prairie, some of these hydrometric stations are operated only seasonally, presenting
additional challenges in using these records for classification exercises (e.g., MacCulloch and
Whitfield, 2012).

As a result, mean annual runoff (Q2) and 1:100 year flood (Q100) magnitudes were

estimated for the 4175 watersheds using relationships defined from canonical correlation
analysis (CCA) to correlate gauged data to multivariate climatic and physiographic data
according to procedures given by Spence and Saso (2005). According to Spence and Saso
(2005), expected uncertainty using these methods approached 50% but exhibited biases of less
than 15% (n = 34). Hydrological stations used were those identified in MacCulloch and
Whitfield (2012) and within the Prairie region (n = 11), and data were obtained from archived
databases of the Water Survey of Canada ([https://wateroffice.ec.gc.ca/search/historical_e.html](https://wateroffice.ec.gc.ca/search/historical_e.html))
between 1990–2014. We note that greater uncertainty than that reported by Spence and Saso
(2005) may result when using the CCA approach with a smaller sample size. Multivariate
geographic data were collected as outlined in the above sections according to the watershed
boundaries for the hydrological stations. Due to the fact that many watersheds within the
HydroSHEDS dataset are likely to drain internally and do not consistently connect to a higher-
order stream network, these streamflow data were interpreted as "runoff", meaning the amount
of water accumulated within the watershed polygon that drains to its lowest point annually.

Briefly, CCA correlates the streamflow record of gauged basins to physico-climatic

characteristics of watersheds by representing these variables as a reduced set of canonical
variables. The analysis results in two canonical variable sets: one for the physico-climatic
variables (i.e., V1 and V2) and another for the hydrological variables (i.e., W1 and W2). These
canonical variables are constructed from linear combinations of the variable sets such that the
correlation of the canonical variables are maximized. Canonical variables plotting similarly on
X-Y plots (W1-W2 and V1-V2), indicate good correlation (Spence and Saso, 2005). Where
canonical correlations ($\lambda_1$, $\lambda_2$) were above 0.75 (Cavadias et al., 2001), that set of physico-
climatic variables was deemed useful for estimating hydrological variables. Those physico-
climatic variables passing this threshold were included as variables in a multiple regression to
develop a predictive equation for Q2. Analyses were performed using the R package *vegan*
(Oksanen et al., 2018).

**3. DATA ANALYSIS**

*3.1. Pre-processing compositional datasets*

Principal components analysis (PCA) was used as a pre-processing step to reduce the

dimensionality associated with compositional datasets (e.g., topographical and land cover
parameters) (Fig. S1). Using this approach, the principal components (PC) that could
cumulatively explain 80% of the variation in a subset of compositional data were included in the
subsequent cluster analysis. This procedure identified the major data patterns and aided in
reducing the number of zero-weighted variables. Where necessary, variables that were not
transformed into PCs were log-transformed to reduce data skewness. Variable unit ranges were
also scaled during the PCA to reduce the impact of certain variables exhibiting a large range of
values on the subsequent cluster analysis.

*3.2. Agglomerative hierarchical clustering of principal components and watershed classification*

Clustering analysis was performed on the suite of physiographic variables, which

included PC variables derived from pre-processing (Table S2; Table S3). Agglomerative
hierarchical clustering of principal components (HCPC) was used to define clusters of
watersheds using the *HCPC* function in the R package *FactoMineR* (Lê et al., 2008; Husson et
al., 2009) to apply a PCA on the standardized multivariate dataset of watershed attributes and
was the basis for clustering. The majority of physiographic variables were included as active
variables in the PCA and thus influenced the arrangements of the PCs. In contrast, watershed
area, DSF, latitude, and longitude were used only as supplementary variables, and thus did not
explicitly affect the clustering analysis. These variables did, however, aid in watershed class
characterization and interpretation. The first set of PCs that together explained 50% of the
variation in the dataset (n = 6) was retained for agglomerative clustering. Retaining these first
PCs at a threshold of 50% allowed for clearer focus on main trends in the data and reduced the
impact of noise on subsequent analyses, which might occur if subsequent, less influential, PCs
were retained.

The agglomerative hierarchical clustering was performed using the Euclidean distances

(from the PCA) and Ward's criterion for aggregating clusters. Ward's criterion decomposes the
total inertia of clusters into between and within-group variance, and this method dictates merging
for clusters (or watersheds) such that the growth in within-group inertia is minimal (Husson et
al., 2010). The total inertia is partitioned into within- and between-group inertias. Within-group
inertia represented the homogeneity, or similarity, of watershed within a cluster. Consequently,
watersheds located close to each other in PC-space were deemed to be similar in their attributes.
Watersheds are grouped according to pairs that minimize within-group inertia (Begou et al.,
2015), and are differentiated based on between-group inertia gained by adding clusters. The
variables contributing to cluster characteristics were determined by v-tests (Husson et al., 2009),
which assessed whether the cluster mean for a given variable was significantly ($p < 0.05$) greater
or smaller than the overall mean.

*3.3. Comparing class-specific observed and simulated wetland depression data*

To compare how well the GPD parameters predicted the observed wetland area

distributions from the Global Surface Water (GSW) dataset, wetland size distributions were
simulated for each class. Wetland area for select watershed class–specific percentiles (i.e., 25th,
50th, and 75th percentiles) derived from the simulated data were then compared to the wetland
areas for corresponding watershed class–specific percentiles of the observed watershed data to
assess the potential usefulness of using these parameters in representing wetland size
distribution.

For this comparison, the fitted wetland area distributions were constrained in their

minimum and maximum values by the Global Surface Water dataset spatial resolution (i.e., the
30 m pixel size) and the median area of the largest wetland observed for each watershed class,
respectively. The median area of the distribution of largest wetlands for each watershed class
gave an indication of the maximum sizes of the water bodies in those watersheds, and thus
provided a maximum value for simulating wetland areas using the GPD. Wetland areas were
simulated using the R package *SpatialExtremes* (Ribatet, 2018).

*3.4. Resampling and re-classifying procedure*
The robustness of the HCPC procedure on characterizing Prairie watersheds was tested
using additional hierarchical clustering on ten subsets of the entire set of 4175. For each
iteration, ten percent of watersheds were removed from the original dataset (n = 4175) without
replacement, and the remaining watersheds (n = 3757) were then re-analyzed according to the
HCPC outlined above (Fig. S1). The number of potential classes allowed was set at seven (k =
7), for consistency with the complete analysis. The resulting classifications were then compared
to the classification performed on the complete dataset, with the watersheds being assessed on
the percentage of iterations in which they were assigned to the same class as the complete
classification. The proportion of membership agreement was calculated and visualized to assess
the likelihood of classing watersheds consistently.

**4. RESULTS**

*4.1. Geographical data processing*
*4.1.1 Dimension reduction: Compositional datasets and principal components analysis*
Variation in geology and soil was best explained by two or three principal components
(Table 1; Fig. S2). Two PCs captured over 80% of the variation in surficial geology, with PC1
(proportion explained: 73%) positively relating to glacial till deposits and negatively with
glaciolacustrine deposits, and PC2 (14%) positively related to riverine or erosive deposits, such
as glaciofluvial, alluvial, and eolian deposits. Particle size class data were explained by the first
two PCs, where PC1 (75%) was positively associated with sand and negatively associated with
silt and clay, while PC2 (14%) was related negatively to silt. Positive PC1 (55%) scores defined
the dominance of black soils, and PC2 (43%) described dominance of brown or dark brown soils
on positive or negative scores, respectively. Three PCs described the local surface form dataset.
PC1 (55%) captured the change from greater portion of hummocky forms to undulating forms,
and PC2 (24%) was negatively associated with higher river-incised landscape fraction. The
portion of level surface form was negatively related to PC3 (12%).

Three PCs were needed to explain over 80% of the variation in land cover (Table 1; Fig.

S2). Land cover PC1 (37%) was positively associated with higher cropland and negatively with
unmanaged grassland; whereas PC2 (25%) was negatively associated with higher pasture and
forest cover. PC3 was associated with greater fallow and pasture areal proportion (21%).
Cropland practice was described by PC1 (90%), with zero-till practices being negatively
associated to this component. Although it only explained 9%, PC2 was also retained to describe
the change between conventional and conservation till practices, with the practices exhibiting a
positive and negative relationship, respectively.

*4.1.2 Canonical correlation analysis*

The canonical coefficients from the CCA were acceptably high at $\lambda_1$ 0.97 and $\lambda_2$ 0.77,

respectively, indicating that the physico-climatic variables exhibited influence on the
hydrological variables (Cavadias et al., 2001; Spence and Saso, 2005). Canonical correlation
values between the hydrological variables and W2 were greater than those with W1 (Table 2);
thus, the physico-climatic variables strongly associated to second canonical correlation (i.e., V2)
were used in the multiple regressions. These variables were watershed area, DSF, areal fraction
of rock, and areal fraction of natural area. Plots of observed and predicted runoff Q2 ($R^2 = 0.45$)
and Q100 ($R^2 = 0.48$) show moderate agreement at lower flow values (Fig. 2). There is a
negative bias estimated between 26 and 29%, which is greater than that documented by Spence
and Saso (2005) using comparable extrapolation methods, but this is not unexpected because of
the smaller sample size in the current study. As Q2 and Q100 exhibited high collinearity, only
Q2 was included in subsequent cluster analyses to:

$$\log(Q2) = 0.130*\log(A) - 0.077*\log(N) + 0.117*\log(R) - 0.141*\log(DSF) - 0.620 \qquad (3)$$


Where *A* was the watershed area, *N* was the natural area fraction and the sum of grasslands and
forest, *R* was the rock fraction area, and *DSF* was the dimensional shape factor of the watershed.
The equation was then used to calculate Q2 for each watershed included in the clustering
analysis.

*4.2. Watershed classification*
*4.2.1. Principal component analysis*

In total, 29 watershed attributes, including the PCs from compositional datasets (see

Table 1), were used in the clustering analysis as active variables, and four were included as
supplementary (Table 3). In the pre-clustering PCA, the first six PCs explained 54.3% of data
variation, and were retained for the HCPC analysis (Fig. 3). The influence of subsequent PCs
declined dramatically, and eleven PCs were required to explain >80%. Variable importance in
the classification was not related to the log-transformed range exhibited by that variable (data not
shown), and impact was mitigated by scaling the ranges of input variables in the PCA.

Principal components 1 and 2 captured changes in physical, land cover, and wetland

characteristics (Fig. 3). PC1 was strongly associated with physical and land cover characteristics,
such as elevation, wetland density, and the land cover PCs. PC2 was strongly related to metrics
characterising the hydrological landscape, including river and wetland density, non-effective area
fraction, landscape surface form, and size of the largest wetland ($W_L$). Subsequent PCs explained
less variation and were more specialized in the variables associated with them. Generally, these
PCs were associated with differences in soil zone and texture class, surficial geology, and
varying surface land form. A more detailed account of associations of the variables with the PCs
is provided below.

PC1 was positively associated with elevation, mean slope, land cover PC2, and surface

form PC3, and negatively with, total annual precipitation, soil zone PC1, wetland density, land
practice PC1, land cover PC1, and longitude (Table 3; Fig. 3). PC2 was associated with non-
effective area fraction, wetland density, *β*, and surface form PC2, and negatively related to land
practice PC1, $W_L$, and river density. PC3 was positively related to wetland fraction, $W_L$, *ξ*, soil
texture PC2, and DSF. Watershed area and runoff were negatively associated with PC3.

Variable correlations were weaker for the remaining three PCs (Table 3). PC4 was

mainly associated with soil texture PC1, surficial geology PC1 and surface landform PC1,
characteristic of sandier soil areas featuring glacial till deposits and higher hummocky surface
forms, as well as higher mean slope. PC4 was negatively related to land cover PC2. PC5 was
related positively to PET, fraction below outlet, and soil zone PC2, and negatively to land cover
PC1, river density, and slope CV. Finally, PC6 was mainly associated with soil texture PC2 and
land cover PC3, and negatively with surface landform PC2.

*4.2.1. Agglomerative hierarchical cluster analysis*
Seven clusters were identified from the hierarchical cluster analysis based on the
between-group inertia gained by increasing cluster number ($k$). The HCPC analysis suggested
three clusters resulted in the greatest reduction of within-group inertia while minimally
increasing $k$ (Fig. 4). Further increasing $k$ refined the separation and differentiation of clusters up
to seven ($k = 7$). Minimal added separation was observed up to $k = 9$, and increasing $k > 9$
resulted in little inertia gained between clusters. Thus, seven clusters, or classes, were manually
selected based on these observations (Fig. 4).

*4.2.3. Class characteristics and interpretation*
Our methodology yields sub-regional watershed classes according to climatic,
physiographic, wetland, and land cover variables. The seven classes (Fig. 5), are defined by
multivariate sets of attributes (Table 4). Influential classifying variables in all classes were mean
elevation, total annual precipitation, land practice, surface forms, and wetland density. Other
variables influential to class differentiation included fraction of non-effective area, land cover,
and soil variables. Climate and elevation gradients are likely responsible for the west to east
watershed clustering pattern. Moreover, we observe strong spatial concordance among some
classes (Fig. 5), which is likely due to the hierarchical nature of the analysis. For simplicity, we
interpret classes based on the variables where large, significant differences in class mean versus
the overall mean of the dataset were observed. The classes can be assigned as follows: Southern
Manitoba (C1); a Prairie Pothole region (C2, C3); Major River Valleys (C4); and Grasslands
(C5, C6, and C7).

*Southern Manitoba (C1)*
The majority of Class 1 (C1; n = 365) watersheds occurred in the eastern prairie south of
Lake Winnipeg (Fig. 5) and thus "Southern Manitoba" is used as the class name. Distinguishing
characteristics associated with this class included soil zone PC1 (predominantly black soils) and
cropland practice PC1 (predominantly conventional till) (Table 4). Southern Manitoba had a high
incidence of glaciolacustrine and alluvial deposits, as indicated by moderately negative and
positive relationships with surficial geology PC1 and PC2, respectively, and the class also had
low mean elevation. Topography tended to be level, with mild slopes and strong association with
land surface form PC3 (Table 4). Notably, these watersheds exhibited both high annual
precipitation and PET compared to other classes, and this class was the only one to have no mean
moisture deficit (i.e., precipitation – PET > 0) (Fig. 6). Southern Manitoba watersheds also
exhibited smaller fractions of non-effective areas and grasslands than other classes (Fig. 7).

*Prairie Potholes (C2 and C3)*
The Prairie Pothole group, consisting of Class 2 (C2; n = 879), or Pothole Till, and Class
3 (C3; n = 681), Pothole Glaciolacustrine, represents the largest class of watersheds spatially,
spanning the northern part of the Alberta prairie to the southeastern part of Saskatchewan (Fig.
5). Mean annual precipitation was relatively high for the study area, contributing to a slightly
negative moisture deficit (Fig. 6). These watersheds contained large fractions of non-effective
area (~75%) (Fig. 7a), and they exhibited positive scores on land cover PC1 (Table 4) indicating
high cropland cover (~70%), whereas unmanaged grassland cover was typically very low
(<20%) (Fig. 7b-c). On average, Pothole watersheds had high wetland densities (wetlands km$^{-2}$),
with C2 exhibiting the greatest density of all classes (Fig. 8a).
Surficial geology differentiated classes C2 and C3. Overall, glacial till and hummocky
landforms dominated the pothole region; however, C2 was more associated with these
characteristics, scoring greater mean values on PC1 of local surface form and surficial geology.
In contrast, glaciolacustrine deposits were more common in C3, and soils had a higher incidence
of clay and silt, where C2 watersheds were sandier (Table 4). Although both classes contain
many wetlands, C2 watersheds had the smallest values of $W_L$, indicating lower areal water extent
was contained in the largest wetland (Fig. 8b).

*Major River Valleys (C4)*
Class 4 (C4; n = 536) watersheds were associated with river valleys, and as such, extend
across the prairie region (Fig. 5) and generally coincide with major rivers (e.g., North and South
Saskatchewan, Qu'Appelle) and large lakes. These watersheds had the greatest value of the
fraction of water area in the largest depression ($W_L$) (Fig. 8b), as well as high slope CV, wetland
fraction, and fractions of black soil (i.e., higher soil zone PC1 scores) (Table 4). These
watersheds were also associated with soil texture PC1 and surficial geology PC2, suggestive of
higher incidence of sandy riverine deposits (e.g., alluvial and glaciofluvial deposits). The Major
River Valleys class tended to have large "wetland" area, which is interpreted as the area of water
of these rivers.

Taken together, these watersheds were related to parameters typical of fluvial

environments, including glaciofluvial or alluvial deposits, and sandier soils. Large values of
mean and CV of slope were also typical of river valley watersheds. About half the basin area
tends to be non-effective in these watersheds, compared to the much greater fractions in the
pothole regions (Fig. 7a) that surround many of the Major River Valleys watersheds. Being river
valleys, C4 watersheds were generally narrow and small in area. Higher DSF (i.e., narrower
watersheds) and smaller areas were generally associated with lower Q2 values (Table 2). Thus,
although these watersheds have a high likelihood of contributing to streamflow of major rivers,
the watershed Q2 contributions were predicted to be small (Table 4).

*Grasslands (C5, C6, and C7)*

The southwestern Canadian Prairie, which includes the majority of southern Alberta and

western Saskatchewan between the South Saskatchewan River and the Cypress Hills, was
occupied by classes C5, C6, and C7. These watersheds tended to have large factions of
unmanaged grasslands (negative land cover PC1) and mean elevation (Table 4). Compared to the
rest of the Prairie, this sub-region tended to be arid, with a strong moisture deficit (Fig. 6). As a
result, these classes exhibited relatively low wetland density (Fig. 8a).

Classes 5 (C5; n = 635), Interior Grasslands, and 6 (C6; n = 702), High-Elevation

Grasslands, were characteristic of the grasslands in southeastern Alberta. These watersheds had
the greatest values of mean fractional grassland area, with cropland and grassland fractions being
comparable (35–40%) (Fig. 7). Distinguishing features of Interior Grasslands were greater values
of the fraction of area below the basin outlet, $A_{BO}$, and a notably large non-effective area fraction
(Fig. 7a). High scores on land cover PC2 and PC3 indicate large fractions of fallow and pasture.
These watersheds also scored higher on soil zone PC2, suggesting more common occurrences of
brown soils. Small magnitudes of mean slope and stream densities were observed, suggesting
that the wetlands within the Interior Grasslands are relatively disconnected from the drainage
network. This characteristic might explain why these watersheds have relatively large wetlands
(Fig. 8c). In contrast, High Elevation Grasslands were characterized by greater mean elevation
and slope values, and smaller non-effective fractions (Table 4; Fig. 7). These watersheds also
had greater stream densities and smaller wetland densities.

Class 7 (C7; n = 377), Sloped Incised, watersheds are characterized by dissected, river-

incised landscapes, as indicated by positive associations with local surface form PC3 (Table 4).
Like High Elevation Grasslands (C6), Sloped Incised watersheds followed the Bow, Red Deer,
as well as the Milk River valleys, suggesting a similar function to those of the Major River
Valleys class. Wetland density is smallest in Sloped Incised watersheds, owing to their steepness,
resulting in surface water reaching stream networks rather than collecting on the landscape (Fig.

8).


*4.3. Predicting wetland size distributions from class parameters*

Simulated wetland area distributions by class were compared to observed size

distributions from study watersheds to evaluate the concordance of the approximate class-
specific distribution to that of the observed distributions of watersheds, collectively. The median
wetland density was greatest in C2, followed by C3, C1, and C5 (Fig. 8a). The median wetland
densities in C6 and C7 were less than 1 km$^{-2}$. C4 had the greatest areal fraction of water in the
largest wetland ($W_L$), which was over 40% (Fig. 8b), while C2 had the smallest value at ~10%.
For the rest of the classes, this value was between 28% and 34%. The simulated wetland area
distributions slightly overestimated those of the observed values, especially at the 25$^{th}$ percentile.
However, the patterns of wetland area in the quartiles was generally consistent among all classes
(Fig. 8c). The area of the smallest 25% of the wetlands appears quite consistent across the
classes, with more variation occurring at higher percentiles. The largest difference among classes
in wetland size was in the 75$^{th}$ percentile, with the greatest range being in C5 and the smallest in
C1.

*4.4. Resampling and re-classifying procedure*

The HCPC and watershed classification was repeated with ten random subsets of 3757

watersheds. The majority of watershed were removed from at least one iteration, with only 50
watersheds being removed a total of 4-6 times (Fig. S3). This resulted in ten unique watershed
subsets to test clustering and agreement to the seven classes, outlined above.

Percent membership agreement of a watershed varied by class, with the majority of

classes exhibiting high agreement even after resampling. Classes exhibiting high membership
agreement were Pothole Till (C2), Interior Grasslands (C5), High Elevation Grasslands (C6), and
Sloped Incised (C7), with a large proportion having more than 90% agreement with the seven
classes from the complete classification (Fig. 9; Table S4). Although a large mean agreement
was observed overall, a few watershed classes exhibited low agreement and inconsistent
classification. Southern Manitoba (C1) exhibited a bimodal distribution, where most were
generally classed as C1 over 75% of the time and a second set only ~60% agreement (Fig. 9).
This was due to a new class appearing (Fig. 10). Hereafter, this class is referred to as "Eastern
Manitoba". Briefly, Eastern Manitoba was association with large fraction of conventional tillage
practice (i.e., positive association with land practice PC1 and land practice PC2) and large
fractional effective areas (data not shown). The Major River Valleys class was the only one that
did not include a watershed that achieved 100% agreement across the ten iterations; this class
exhibited a peak of total agreement at approximately 60% (Fig. 9). Where Major River Valleys
watersheds were classified inconsistently, the most common alternative classification were
Pothole Glaciolacustrine (C3) or secondarily High Elevation Grasslands (C6) (Fig. 10). The loss
of Major River Valleys occurred for iterations when the Eastern Manitoba class (C8) became
apparent.

**5. DISCUSSION**

*5.1. Classifying Prairie watersheds*
*5.1.1. Hydrological approaches*

Our classification procedure grouped watersheds of approximately 100 km$^2$ into seven

classes. Few studies anywhere have classified watersheds at this granularity, and our
investigation gives particular attention to characteristics that influence hydrological and
ecological behaviour. Many previous studies in the region spanned larger areas, and this often
results in the Prairie being identified as a homogenous region due to relatively low streamflow
and atypical geology and surface topography (MacCulloch and Whitfield, 2012; Mwale et al.,
2011). Our results are novel in that they characterize in greater detail, and at small watershed
scales, the potential for different hydrological behaviour of watersheds within the region. The
only similar example that was found in the literature was by Durrant and Blackwell (1959),
whose findings parallel those of this study, but at a larger watershed scale. Durrant and
Blackwell (1959) described broad regions of Saskatchewan and Manitoba based on mean annual
flood, distinguishing five sub-regions including southwestern Saskatchewan, north and central
Saskatchewan, and southern Manitoba near the Red River and Assiniboine River confluence. In
the current study, surficial geology and land surface form strongly influenced how grasslands
were separated into three classes, which reinforces the role of local topography on hydrological
response, as seen elsewhere (Mwale et al., 2011). Likewise, surficial geology was particularly
important for distinguishing the Pothole (Till and Glaciolacustrine) classes. Similarities to the
work of Durrant and Blackwell (1959) based on streamflow in larger basins suggest that our
approach, with consideration of factors important to watershed behaviour, can yield
classification with relevance to hydrologic function, despite the use of few hydrologic indices in
our analysis (Fig. 5). This approach holds potential for use in other regions of the world that are
dry, ungauged, or feature low effective areas, and thus cannot rely on streamflow characteristics
as a primary means of classification according to functional behaviour.
Our classification grouped Prairie watersheds using geological, biophysical, and
hydroclimatic attributes. In their review of classification approaches, Sivakumar et al. (2013)
indicate that solely using geographic data is advantageous when there are limited hydrological
data; however, the relationship between physical attributes and hydrologic behaviour is not
necessarily definitive in all regions. For these reasons, it was important to include traits
indicative of structural hydrological connectivity, such as Q2 estimates and wetland parameters.
It is important to note that while Q2 emerged as a defining feature for several of the classes, it
was consistently one of many variables important for characterization of that class (Table 4),
suggesting that while it provides value added, it does not stand out as a major driving factor in
the classification. In particular, the immature drainage network and relatively high depressional
water storage capacity make prairie hydrology relatively distinct (Jones et al., 2014; Shook et al.,
2013, 2015). Notably, three classes (i.e., Pothole Till, Pothole Glaciolacustrine, and Interior
Grasslands) occur almost exclusively within regions that tend not to contribute to major river
systems, and collectively encompass the majority of the study area (Table 4; Fig. 5). It is
therefore expected that hydrological response will be very different between classes that exhibit
higher hydrological connectivity (i.e., potentially lower wetland to stream densities and non-
effective area fractions), such as the Major River Valleys or Sloped Incised watersheds, than
those that do not, such as Pothole classes.

*5.1.2. Ecoregions and human impacts*
Ecoregions are commonly used to characterize landscapes according to geographical or
ecological similarity (Omernik and Griffith, 2014). Similar to our approach, ecoregion
classifications are often hierarchical in nature, allowing for differing levels of detail, spatial
extent, and thus defining characteristics depending on the scale of interest (Loveland and
Merchant, 2004). Ecoregion classifications used in the United States (Omernik and Griffith,
2014) and Canada (Ecological Stratification Group, 1995) employ a "top-down" approach,
where broad categories are partitioned into smaller, more specialized units. In contrast, our
approach provides a bottom-up, agglomerative approach where similar watersheds are merged.
Assumptions are inherent in either approach; however, the latter was applicable to the current
study to allow for grouping of watersheds given similarities in physiographic characteristics.
This approach does not limit class membership to the geographic extent of a higher level class,
allowing for membership to potentially span the geographic extent of the Canadian Prairie
domain (Fig. 5).
Despite the differing methods for distinguishing similarities (or differences),
arrangements of watershed classes in some cases exhibited similar ranges to ecoregion
boundaries. The boundaries of Lake Manitoba Plain and Mixed Grassland ecoregions
(Ecological Working Group, 1995) correspond roughly to those of the broader Southern
Manitoba (C1) and Grasslands (C5, C6, and C7) classes, respectively (Fig. S4). Mwale et al.
(2011) also found that annual hydrological regimes based on data from 200 stations and physical
attributes in Alberta linked closely with provincial ecoregions. Our emphasis on inclusion of
hydrologically relevant characteristics, such as wetland traits and effective areas that are likely
important contributors to function, has proven useful for further distinguishing among the
Grassland classes as well as the Pothole classes (C2 and C3) (Fig. 5; Fig. S4). Due to the
fundamental differences in effective areas and in wetland versus river dominated systems (Table
4; Fig. 8), we expect different hydrological behaviour between these classes. This is an
advantage of the HCPC classification approach in that it allows for identifying the potential
similarity at relatively fine spatial scales, and does not require similar watersheds to be
physically adjacent to one another. This confers the opportunity to further investigate these
systems, such as through hydrological modelling and contrasting resulting responses under
climate and land-use scenarios.
The highly managed prairie landscape reinforces the importance of considering
anthropogenic alteration in hydrological understanding. Crop rotation and the ways in which
fields are managed for winter affect the accumulation and redistribution of snow (Fang et al.,
2010; Harder et al., 2018; Van der Kamp et al., 2003). Spring snowmelt and consequent runoff
are imperative to summer surface water availability (Dumanski et al., 2015; Shook et al., 2015),
and depression-focused recharge of snowmelt into groundwater facilitates storage and mitigates
flood impacts (Hayashi et al., 2016). Thus, classifying procedures in the Prairie must consider
the human influence on the water cycle.
An example of the complexities introduced by human land management activities can be
shown by the C1 (Southern Manitoba) watersheds, where the land practice variable was a strong
class descriptor. Agricultural activity is high everywhere in the Prairie; however, only C1 was
associated with low zero-till practices, instead favouring conventional tillage (Table 4).
Manitoba has seen less coherent adoption of zero-till practices since the early 1990s in compared
to Alberta and Saskatchewan, with conventional or other conservation till practices remaining
common in Manitoba (reviewed in Awada et al., 2014). Sustained use of conventional tillage
practice within this region may increase the risk of soil erosion, which can negatively affect
downstream water bodies (Cade-Menun et al., 2013). This practice, combined with landscape
modifications, such as artificial drainage networks, serve to facilitate removal of water and may
contribute to concurrent nutrient export from agricultural lands (Weber et al., 2017).
These management practices can be viewed as a trade-off, where high numbers of
wetlands and level topography can pose flood risk during wet periods as wetlands fill and merge
(Leibowitz et al., 2016), inundating tracts of adjacent land. Conversely, where landscape
modification to enhance water export occurs, local, field-scale flood risk may be reduced, while
heightening the risk of downstream flooding. Land-use and land management are important
factors in understanding the connectivity and chemical transport in prairie landscapes (Leibowitz
et al., 2018). In southern Manitoba, where artificial drainage has been used to increase the area of
arable land, beneficial management practices in the form of agricultural reservoirs have been
implemented as a means of reducing nutrient export and improving downstream water quality
while also mitigating the risk of downstream flooding (Gooding and Baulch, 2017). These
factors illustrate the complexities when classifying and understanding hydrological response of
watershed embedded in highly managed landscapes, and underscore that necessity of considering
the human influence on the water cycle in such approaches.

*5.2. HCPC as a clustering and classification framework*
*5.2.1. Using the HCPC approach and limitations*
The HCPC method provides a procedure for integrating multiple physiographic attributes
and describes resulting clusters by sets of significant variables (Husson et al., 2009). As
discussed above, an advantage of the method is that it groups individual watersheds based on
similarities. Therefore, it lends itself well as a foundation for investigating hydrological
behaviour through modelling efforts. In the case of the current study, modelling efforts can be
applied at a 100 km$^2$ scale to evaluate responses to environmental changes. An additional
advantage is that that one may select variables or sets of variables of interest to inform the
clustering of watersheds, such as those based only on topographic parameters or those dictating
local hydrology. For example, climate variables may be excluded if the goal of the classification
is parameterizing a hydrological model, as these variables could instead be described by local
climate forcing. The relative ease with which different sets of variables can be added to or
excluded from the analysis to consider different permutations of the classification is a real
strength of the approach. Although this may result in differing cluster results, assessment of how
these classes change with addition or removal of certain datasets can identify the variables that
control class definition as well as elucidate spatial patterning of classes.
There are a few considerations when using this method. First, the linear restrictions of
this method are challenging when working with environmental data, which often do not conform
to assumptions of normality. Non-linear PCA methods and self-organizing maps have been
applied successfully to classify watersheds in Ontario and to regionalize streamflow metrics
(Razavi and Coulibaly, 2013, 2017). Although these methods might be logical next steps for the
current study, we chose to focus on conventional PCA due to its smaller computational cost
when classifying the large number of watersheds in our study.

Second, the current analysis weighs all variables equally. This can bias the analysis

towards attributes that exhibit greater variability, as these can overshadow other more
constrained variables. For example, the location of the largest pond relative to the watershed
outlet (coded as $L_W/L_O$) is important to controlling local prairie hydrology and hydrological gate-
keeping potential (i.e., the likelihood of releasing surface water to the next order watershed)
(Shook et al., 2013, 2015) and water quality (Hansen et al., 2018). Despite its hydrological
importance, this variable had little influence on the clustering procedure overall, and was only a
minor descriptor in certain classes, such as C5 and C6 (Table 4).

The original set of watersheds in the clustering analysis can affect the final classification;

however, there was a high degree of agreement between classified subsets of the original dataset,
and the classification generated using the complete set of watersheds (n = 4175) (Fig. 9). Overall,
watersheds designated as part of the Pothole and Grassland classes were classified consistently,
with most exhibiting over 90% agreement. Major River Valleys exhibited the weakest agreement
(Fig. 9), due to the appearance of a unique (new) class consistent with the Lake Manitoba Plain
ecoregion (Fig. S4) for some of the subsets. In these cases, those watersheds previously
classified as Major River Valleys were re-distributed to mainly High Elevation Grasslands or
Pothole classes depending on dominate watershed features (Fig. 10). Although we do not include
a detailed account of the new Eastern Manitoba class that emerged during this exercise, defining
characteristics included a high fraction of effective area (i.e., the most eastern portion of the
Prairie in Fig. 1), low relief, and lower use of zero-till agriculture (as reviewed in Awada et al.,
2014). Since this new class would not be expected to translate to notable differences in
management outcomes. Moreover, previous reviews on the usefulness of ecoregion
classifications agree that strict geographic boundaries are unlikely, and are instead more likely
"fuzzy" (Loveland and Merchant, 2004; Omernik and Griffiths, 2014).

Class membership in our approach is also determinate. In reality, there can be large

variability in attributes within a class (e.g., Fig. 7), and membership is determined by the
collective similarity of watershed attributes. Previous studies have used fuzzy c-means and
Bayesian approaches that can assign a likelihood of membership to classes (Jones et al., 2014;
Rao and Srinivas, 2006; Sawicz et al., 2011). An advantage to this approach is that it allows for
fuzzy boundaries between classes where a gradient of features likely exists (Loveland and
Merchant, 2004). Our re-classifying analysis supports the proposition that boundaries among
classified regions are fuzzy and some watershed might flicker among class memberships (Fig.
10). Such approaches are also un-supervised and probabilistic in nature and will eliminate the
subjectivity due to the researcher pre-defining the number of classes. Future work thus should
consider these fuzzy boundaries and potential for watersheds to exhibit partial membership to
multiple classes.

*5.2.2. Data quality and availability*
The classes resulting from the HCPC are also ultimately dependent on the types of data
included. The availability of data and its geographic coverage determined the environmental
parameters included in our analyses. Ideally, a more detailed estimate of runoff for each
watershed would be a valuable contribution. In the current study, we used the CCA and eleven
reference stations to approximate runoff values for the clustering watersheds. Given the number
of watersheds included in the analyses, the diversity of physical characteristics and potential
hydrological behaviour is likely not completely represented in the small sample size of available
hydrometric stations, and is a limitation of our approach. Soil moisture would be important to
consider in future studies given its role in influencing vegetation community composition, PET,
and over all water balance (Hayashi et al., 2003; Shook et al., 2015). Where data is available,
future work should consider variables related to snow formation and melt, as well the proportion
of annual precipitation as snowfall. These variables are likely influential when describing
hydrological behaviour of the watersheds and classes in the current study, and other cold regions
(Knoben et al., 2018; Shook and Pomeroy, 2012). Furthermore, a comprehensive wetland
inventory or an index of wetland drainage activity that is comparable across the three Provinces
does not currently exist. These would be valuable additions to future efforts to classify Prairie
watersheds given the important role of land modification on watershed functions.
One consideration with the Global Surface Water dataset is that the pixel size (30 m) is
quite coarse and will miss numerous smaller wetlands, underestimating the number of wetlands
observed. Consequently, it is likely that the analysis omitted some ephemeral wetlands for which
persistence is short and size is small. Despite their known important ecological functions
(Calhoun et al., 2017; Van Meter and Basu, 2015), their size and transient nature is a challenge
to their inclusion in comprehensive datasets spanning large geographic areas. This may
inadvertently result in the role of smaller wetlands being under-represented in our analysis, or
others that rely on this dataset.
Use of the $\xi$ and $\beta$ parameters as indices of the wetland area frequency distributions were
shown to estimate classes area distributions reasonably well (Fig. 8c). Although for consistency,
we restricted our simulated dataset to the spatial resolution of the surface water raster, one could
use these parameters to estimate the frequencies of smaller wetlands in watersheds, which would
otherwise be missed by satellite measurements, assuming conformity to a Generalized Pareto
Distribution (Shook et al., 2013). Our analysis supports this application as simulated wetland
areas generally approximated those seen across the observed data (Fig. 8c). Nonetheless, in
regions where wetland drainage has been undertaken, it is expected that wetland area distribution
has been altered via preferential loss of smaller water bodies (Evenson et al., 2018; Van Meter
and Basu, 2015). This is exacerbated by the fact that remote sensed satellite data tends to omit
smaller, ephemeral ponds. A more robust characterization of the size and permanence of
wetlands in our study watersheds would be expected to improve the current dataset and enhance
the clustering and classification analyses.

*5.3. Management implications*
Classification frameworks help to define sub-regions with potentially similar
characteristics or behaviours. For example, climatic zones can be delineated, specifically the dry
Grassland watersheds in the southwest and the wet Potholes in the northeast and in Manitoba
(Fig. 5). In some cases, this may be related to local wetland densities, with large densities
observed corresponding with low moisture deficits (Fig. 6b) (Liu and Schwartz, 2012). Climate
variation may divide watersheds with seemingly similar geography into differing classes, as is
the case with Major River Valleys and Sloped Incised watersheds. Both sets of watersheds
tended to follow river valleys, but the former exhibit greater precipitation and smaller PET
(Table 4). These divisions can be used to give context to regions we might expect to behave
similarly, whether hydrologically, or ecologically, based solely on physical attributes, and echoes
other methods, such as ecodistricts (Ecological Stratification Working Group, 1995) to classify
landscapes. For example, areas that are geologically similar may differ in terrestrial or aquatic
community assemblages, which should influence how each area might be managed (Jones et al.,
2014; Wagner et al., 2007). If classifications are used to inform management, the resulting
decisions for a given location will depend on the strength of the delineation, the scale at which
management is applied, relationships among management practices and the attributes used to
define that area, and the relationship of those attributes to the response variable of concern
(Wagner et al., 2007).

This set of analyses was unique among watershed classification exercises in Canada in

that it considered a suite of wetland variables. The arrangement of wetlands or landscape
depressions and their size distribution define the hydrological behavior of Prairie watersheds
(Shook et al., 2015; Shook and Pomeroy, 2011). The storage capacity and subsequent spilling or
merging controls wetland connectivity, and thus the quantity of water available to move from
one watershed to another (Leibowitz et al., 2016; Shaw et al., 2012; Shook et al., 2015). In turn,
a wetland or depression's hydrological gate-keeping potential, or its likelihood to prevent
connectivity to the downstream watershed, is a function of both its storage capacity and
landscape position. Large wetlands near an outlet have a great gate-keeping potential, as they
block much of the watershed from connecting, and it takes a great deal of water to fill them
before permitting flow to the next order watershed (Shook and Pomeroy, 2011). Simulated
frequency distributions of wetland areas indicate that the depressional storages of the classes are
very different (Fig. 8). It may be that wetland management practices will have different
influences between each pothole class, and possibly among all the classes. This has implications
for managing salinizing soils (Goldhaber et al., 2014), biodiversity (Balas et al., 2012), and
flooding potential (Evenson et al., 2018; Golden et al., 2017).

Wetland drainage and wetland consolidation change hydrological connectivity and

therefore the transport of nutrients and their loading into receiving water bodies (Brown et al.,
2017; Vanderhoof et al., 2017). More positive values of the moisture deficit (i.e., where P >=
PET) were associated with greater wetland densities (Fig. 6b) (Liu and Schwartz, 2012), and
these areas were generally associated with greater fractions of cropland, such as Pothole Till,
Pothole Glaciolacustrine, and Southern Manitoba watersheds. In these regions wetland drainage
is widely practiced, historically or at present, and conflict over available arable land and wetland
conservation is high (Breen et al., 2018).

Extensive drainage in combination with agricultural activity is known to increase the risk

of agricultural nutrient mobility (Kerr, 2017) from the landscape to receiving water bodies.
Increased connectivity also reduces water residence time and thus tends to decrease wetland
nutrient retention (Marton et al., 2015). Over time, zero-till practices can promote nutrient
stratification in soils, where concentrations (especially phosphorus) accumulate at the surface,
which can increase nutrient loading when surface runoff is generated (Cade-Menun et al., 2013).
The cropland-wetland interface might also have important implications for pesticide mobility in
Pothole Till and northern Pothole Glaciolacustrine watersheds. These areas coincide with
extensive use of canola, which has been linked to high application rates of neonicotinoid
pesticides which are known to have high persistence in small, temporary wetlands (Main et al.,
2014). Watersheds in the Pothole Till class appear to have more hummocky landscapes than the
Pothole Glaciolacustrine classification and smaller, more numerous wetlands (Fig. 8). Moreover,
the water area fraction occupied by the largest wetland differs between the classes. The
landscape biogeochemical functionality of pothole wetlands is known to vary considerably
according to pothole character (Evenson et al., 2018; Van Meter and Basu, 2015). As such, our
classification may highlight contrasting biogeochemical functioning, including nutrient retention,
between these classes. Thus, although water quality risks are common within the region, the
classes may respond very differently to environmental and land management stresses.

**6. CONCLUSION**

This study provides an overview of a classification framework that can be applied in
regions with limited understanding of or data describing streamflow. The HCPC procedure offers
a flexible analysis to elucidate the spatial arrangement of watershed classes given a large number
of units to classify and a diverse set of attributes to inform the classification. In contrast to
classifications based solely on hydrological function, using physiographic data allows for
classifying small basins, which are unlikely to be gauged, and confers advantages over alternate
procedures that rely heavily on observations of hydrological parameters, namely statistics
describing streamflow.
Use of the classification approach for small Canadian Prairie watersheds identified
regions of similar climatic and geographic features and, potentially, of hydrological response
(Fig. 5). This yielded watershed classes that consider not only drainage patterns, but also land
cover, land-use, and the underlying geology. In the Prairie region, wetland variables incorporate
the hydrologic gate-keeping potential of wetlands as well as parameters indicative of wetland
size distributions. With the classification based on a large and diverse set of attributes, a diversity
of behaviours is captured. This represents a major step forward for classification of Prairie
watersheds that have to-date offered only a much more homogenized depiction of watershed
function in the region. The watershed classification framework presented promises to be useful
in other dry or semi-arid regions, and those that are poorly gauged. Given the inclusive nature of
the classification approach, which incorporates landscape controls on hydrology as well as those
influencing biogeochemistry and ecology, it also provides a foundation to evaluate the efficacy
of land and watershed management practices in the context of a changing climate.







**Author contributions**
JDW, CJW, and CS conceived the study, and JDW collected data and performed analyses. KRS
wrote code to analyze basin and wetland data. JDW wrote the manuscript with input from all co-
authors.

**Acknowledgements**
This work was pursued under the Prairie Water project and funded by the Global Water Futures
program, which was supported by the Canada First Research Excellence Fund. The authors
would like to thank John Pomeroy for his valuable input on the scoping and approach to the
study. We would also like to thank Wouter Knoben and two anonymous reviewers for their
insightful comments on the manuscript. Finally, we would like to thank the Prairie Water team
and the Global Institute for Water Security for ongoing support. The authors declare that they
have no conflict of interest.

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

 **TABLES AND FIGURES**

**Table 1 –** Pre-processing of compositional data PCA results. Shown are the respective subsets,
the number of initial fractional area variables before dimensional reduction, the number of
principal components retains to reach over 80% of subset variation (except for tillage practice),
and the proportion of variation explained by each component.

| Variable subset | Number of initial variables | Number of principal components | Total variation explained by component |
|---|---|---|---|
| Surficial geology | 6 | 2 | 1: 72.8%<br>2: 14.4% |
| Particle size class | 3 | 2 | 1: 74.8%<br>2: 15.6% |
| Soil zone | 5 | 2 | 1: 54.6%<br>2: 42.7% |
| Local surface form | 5 | 3 | 1: 54.5%<br>2: 24.2%<br>3: 11.9% |
| Land cover | 5 | 3 | 1: 36.8%<br>2: 25.2%<br>3: 20.6% |
| Tillage practice | 3 | 2 | 1: 90.9%<br>2: 8.5% |


**Table 2 –** Canonical correlation coefficients for watershed attribute and hydrological variables of
hydrological research stations from the canonical correlation analysis. Those variables used in
multiple regression equations are denoted with a '*'.

|  | Correlation | |
| --- | --- | --- |
| **Watershed attributes** | **V1** | **V2** |
| Area* | 0.36 | –0.83 |
| DSF* | –0.26 | 0.90 |
| Fraction rock* | –0.64 | 0.61 |
| Fraction natural area* | –0.26 | 0.71 |
| Stream density | –0.27 | 0.37 |
| Mean annual precipitation | –0.14 | –0.30 |
| Fraction water area | 0.53 | –0.19 |
| **Hydrological variables** | **W1** | **W2** |
| Q2 | –0.82 | –0.58 |
| Q100 | –0.22 | –0.98 |
| Canonical $\lambda$ | 0.97 | 0.77 |



**Table 3** – Correlation of study watershed attributes to principal components (PC). The values for
the six PCs used in the cluster analysis are shown.

| Variable | Abbreviation | PC1 | PC2 | PC3 | PC4 | PC5 | PC6 |
|---|---|---|---|---|---|---|---|
| Mean elevation | elevation | 0.81 | 0.34 | –0.14 | 0.09 | –0.16 | –0.17 |
| Mean slope | slope.mean | 0.61 | –0.23 | 0.06 | 0.37 | –0.10 | 0.11 |
| Slope CV | slope.CV | 0.30 | –0.38 | 0.22 | 0.14 | –0.41 | –0.09 |
| Total precipitation | precip | –0.85 | –0.12 | 0.13 | 0.16 | –0.07 | 0.30 |
| Potential evapotranspiration | PET | 0.31 | –0.33 | –0.33 | –0.06 | 0.47 | 0.13 |
| Non-effective area | NE.area | 0.02 | 0.70 | 0.31 | 0.10 | 0.01 | –0.15 |
| Areal fraction below outlet ($A_{BO}$) | A_BO | 0.14 | 0.25 | 0.27 | –0.17 | 0.42 | –0.01 |
| Stream density | stream.density | 0.08 | –0.42 | –0.39 | 0.03 | –0.41 | 0.08 |
| Wetland density | wetland.density | –0.63 | 0.46 | 0.11 | –0.04 | 0.12 | 0.24 |
| Wetland fraction | wetland.frac | –0.30 | 0.19 | 0.66 | –0.36 | 0.02 | 0.11 |
| Water area in largest wetland to total in watershed ($W_L$) | W_L | 0.31 | –0.44 | 0.51 | –0.32 | –0.06 | –0.12 |
| Location of largest wetland to outlet ($L_W/L_O$) | L_W/L_O | –0.01 | –0.06 | –0.22 | 0.09 | –0.07 | –0.07 |
| Beta ($\beta$) | beta | 0.17 | 0.49 | –0.02 | 0.01 | 0.09 | 0.05 |
| Xi ($\xi$) | xi | 0.21 | –0.23 | 0.57 | –0.31 | –0.10 | –0.17 |
| Runoff (Q2) | Q2 | –0.13 | 0.35 | –0.47 | 0.00 | –0.33 | 0.10 |
| Soil texture PC1 | Text.PC1 | –0.07 | –0.04 | 0.28 | 0.55 | 0.19 | –0.32 |
| Soil texture PC2 | Text.PC2 | 0.02 | –0.32 | 0.43 | 0.03 | –0.31 | 0.54 |
| Soil zone PC1 | Soil.PC1 | –0.65 | –0.29 | –0.07 | 0.19 | –0.10 | –0.24 |
| Soil zone PC2 | Soil.PC2 | 0.27 | –0.12 | –0.06 | –0.11 | 0.40 | 0.25 |
| Land cover PC1 | LC.PC1 | –0.44 | 0.38 | –0.21 | –0.26 | –0.43 | 0.12 |
| Land cover PC2 | LC.PC2 | 0.42 | 0.22 | –0.17 | –0.53 | 0.15 | 0.03 |
| Land cover PC3 | LC.PC3 | 0.21 | 0.30 | 0.15 | 0.25 | 0.11 | 0.46 |
| Surficial geology PC1 | SF.PC1 | 0.06 | 0.21 | –0.19 | 0.50 | 0.17 | –0.09 |
| Surficial geology PC2 | SF.PC2 | 0.06 | –0.38 | 0.24 | 0.47 | 0.11 | –0.03 |
| Surface form PC1 | LL.PC1 | –0.16 | 0.20 | 0.17 | 0.47 | 0.26 | 0.26 |
| Surface form PC2 | LL.PC2 | –0.20 | 0.44 | 0.12 | –0.03 | 0.04 | –0.55 |
| Surface form PC3 | LL.PC3 | 0.41 | 0.38 | 0.20 | 0.21 | –0.27 | 0.27 |
| Land practice PC1 | LP.PC1 | –0.54 | –0.58 | –0.13 | –0.10 | 0.32 | –0.09 |
| Land practice PC2 | LP.PC2 | 0.14 | –0.16 | –0.24 | –0.22 | 0.29 | 0.30 |
| *Supplementary variables* | | | | | | | |
| Latitude | Lat | –0.15 | 0.24 | 0.26 | –0.01 | –0.33 | –0.41 |
| Longitude | Long | –0.73 | –0.24 | 0.06 | 0.10 | 0.16 | 0.39 |
| Area | Area | –0.05 | 0.27 | –0.44 | 0.09 | –0.15 | –0.03 |
| DSF | DSF | –0.02 | –0.25 | 0.42 | –0.05 | 0.12 | 0.01 |


 **Table 4 –** Classes and distinguishing variables of prairie watersheds. The v-test statistics, based
on Ward's criterion, are shown. Variables with v-test values greater or less than 10 and –10,
respectively, are bolded to emphasize defining features of each class. All variables are significant
to p < 0.001. *Classes: Southern Manitoba (1), Pothole Till (2), Pothole Glaciolacustrine (3),*
*Major River Valleys (4), Interior Grasslands (5), High Elevation Grasslands (6), Sloped Incised*
*(7).*

| Class 1 (n=365) | | Class 2 (n=879) | | Class 3 (n=681) | | Class 4 (n=536) | |
|---|---|---|---|---|---|---|---|
| Variable | v-test | Variable | v-test | Variable | v-test | Variable | v-test |
| **LP.PC1** | **48.11** | **wetland.density** | **28.23** | **LC.PC1** | **22.60** | **SF.PC2** | **19.83** |
| **precip** | **30.33** | **LL.PC1** | **24.81** | **wetland.frac** | **12.74** | **slope.CV** | **19.35** |
| **Soil.PC1** | **23.60** | **precip** | **22.74** | **Q2** | **12.63** | **xi** | **16.05** |
| **LP.PC2** | **14.74** | **SF.PC1** | **21.74** | **NE.area** | **11.12** | **W_L** | **15.39** |
| **PET** | **13.10** | **LC.PC1** | **17.19** | LL.PC2 | 9.45 | **Text.PC2** | **15.07** |
| wetland.density | 7.39 | **LL.PC2** | **16.42** | wetland.density | 8.05 | **Text.PC1** | **14.40** |
| DSF | 6.81 | **Q2** | **15.77** | LC.PC2 | 6.70 | **Soil.PC1** | **14.01** |
| SF.PC2 | 6.53 | **Soil.PC1** | **15.76** | LL.PC3 | 6.53 | **DSF** | **11.76** |
| stream.density | 4.61 | **NE.area** | **15.72** | xi | 5.89 | **precip** | **10.97** |
| LC.PC1 | –3.37 | **area** | **13.15** | W_L | 4.58 | **wetland.frac** | **10.92** |
| A_BO | –4.22 | **Text.PC1** | **12.00** | precip | 3.47 | slope.mean | 7.29 |
| area | –5.46 | LC.PC3 | 6.76 | A_BO | –3.79 | LP.PC1 | 3.52 |
| slope.CV | –6.49 | beta | 5.31 | slope.CV | –4.97 | A_BO | –3.83 |
| Q2 | –8.47 | L_W/L_O | 4.20 | L_W/L_O | –5.17 | wetland.density | –4.41 |
| SF.PC1 | –8.90 | LL.PC3 | 3.93 | LP.PC2 | –7.11 | SF.PC1 | –4.56 |
| LC.PC2 | –9.21 | SF.PC2 | –3.97 | LC.PC3 | –9.71 | LC.PC1 | –5.13 |
| **LL.PC2** | **–14.18** | LP.PC1 | –4.87 | **LP.PC1** | **–12.38** | soil.PC2 | –6.93 |
| **slope.mean** | **–16.17** | stream.density | –5.92 | **Soil.PC2** | **–13.01** | beta | –7.60 |
| **beta** | **–16.88** | elevation | –7.15 | **Text.PC1** | **–14.58** | elevation | –8.03 |
| **LC.PC3** | **–18.13** | A_BO | –7.86 | **slope.mean** | **–15.92** | **area** | **–11.04** |
| **NE.area** | **–28.97** | Text.PC2 | –9.15 | **SF.PC2** | **–17.03** | **LP.PC2** | **–11.44** |
| **LL.PC3** | **–36.59** | DSF | –9.93 | **LL.PC1** | **–17.83** | **Q2** | **–13.27** |
| **elevation** | **–47.42** | **LP.PC2** | **–10.88** | **SF.PC1** | **–18.83** | **PET** | **–13.98** |
| | | **Soil.PC2** | **–12.00** | **PET** | **–23.29** | **LC.PC2** | **–20.86** |
| | | **PET** | **–13.15** | | | | |
| | | **slope.mean** | **–13.50** | | | | |
| | | **slope.CV** | **–16.26** | | | | |
| | | **LC.PC2** | **–16.29** | | | | |
| | | **xi** | **–21.49** | | | | |
| | | **W_L** | **–32.96** | | | | |


 **Table 4** – *(cont'd)*

| Class 5 (n=635) | | Class 6 (n=702) | | Class 7 (n=377) | |
|---|---|---|---|---|---|
| Variable | v-test | Variable | v-test | Variable | v-test |
| **A_BO** | **34.10** | **elevation** | **29.29** | **Text.PC2** | **27.65** |
| **LC.PC2** | **21.53** | **PET** | **20.16** | **LL.PC3** | **25.69** |
| **Soil.PC2** | **20.81** | **slope.CV** | **17.67** | **slope.mean** | **22.32** |
| **LC.PC3** | **17.44** | **slope.mean** | **16.12** | **LC.PC3** | **14.84** |
| **NE.area** | **16.22** | **stream.density** | **14.55** | **stream.density** | **13.82** |
| **beta** | **15.96** | **LC.PC2** | **14.09** | **Soil.PC2** | **13.09** |
| **elevation** | **13.31** | W_L | 9.47 | **elevation** | **12.42** |
| **PET** | **11.47** | L_W/L_O | 6.80 | **PET** | **11.47** |
| LL.PC2 | 8.11 | LP.PC2 | 5.73 | SF.PC2 | 6.80 |
| LP.PC2 | 7.67 | area | 3.72 | LP.PC2 | 6.39 |
| LL.PC3 | 7.31 | LL.PC2 | 3.62 | slope.CV | 5.87 |
| wetland.frac | 5.77 | LP.PC1 | –3.60 | W_L | 4.63 |
| LL.PC1 | 5.50 | Q2 | –3.94 | precip | –4.75 |
| SF.PC2 | –4.74 | DSF | –4.91 | A_BO | –5.65 |
| area | –4.86 | A_BO | –9.47 | LC.PC1 | –7.62 |
| L_W/L_O | –7.11 | **Soil.PC1** | **–10.17** | Text.PC1 | –8.34 |
| Q2 | –9.34 | **LL.PC3** | **–10.62** | **LP.PC1** | **–11.42** |
| LP.PC1 | –9.96 | **LC.PC3** | **–13.17** | **NE.area** | **–13.33** |
| **Text.PC2** | **–11.36** | **NE.area** | **–14.11** | **wetland.frac** | **–13.64** |
| **LC.PC1** | **–11.38** | **LL.PC1** | **–15.44** | **wetland.density** | **–16.27** |
| **slope.CV** | **–12.42** | **Text.PC2** | **–15.78** | **Soil.PC1** | **–16.43** |
| **precip** | **–20.86** | **LC.PC1** | **–17.15** | **LL.PC2** | **–39.41** |
| **Soil.PC1** | **–23.58** | **wetland.frac** | **–21.48** | | |
| **stream.density** | **–26.34** | **wetland.density** | **–29.58** | | |
| | | **precip** | **–37.27** | | |


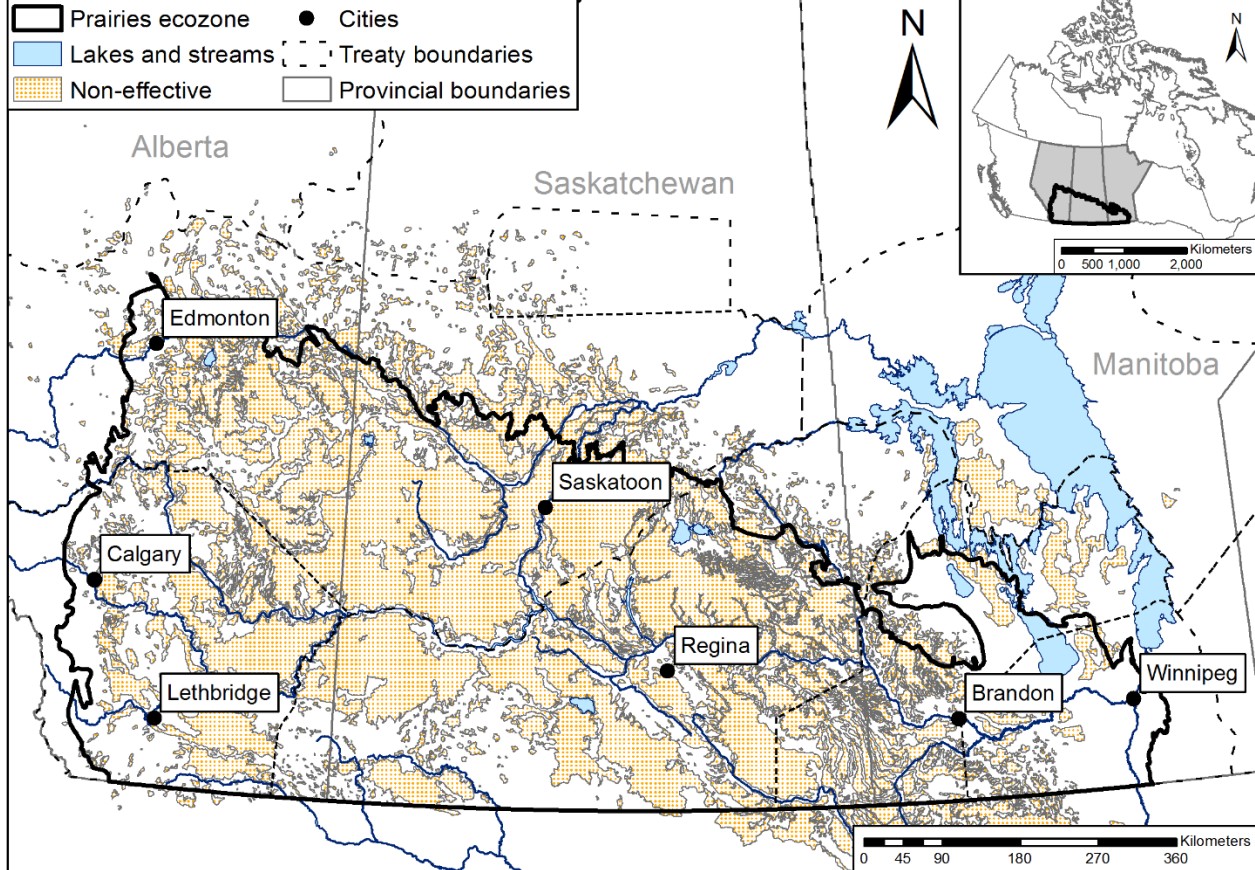


**Figure 1 –** Map of the study area spanning the Prairies ecozone in western Canada (inset). Large
cities in each of the three provinces are shown for reference, while the region characterized as
not contributing runoff (2-year) is also shown. Prairie ecozone based on the region classified by
the Ecological Stratification Group (1995).


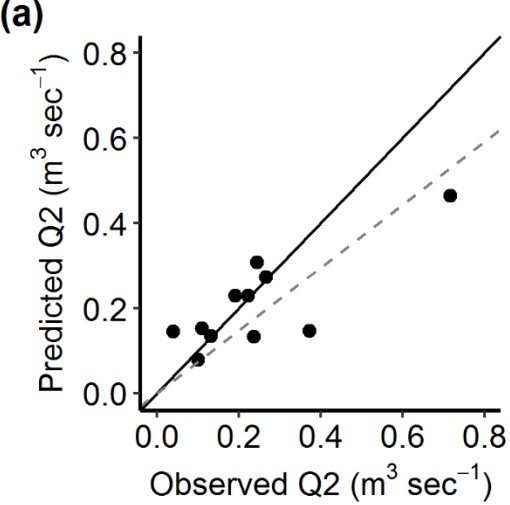

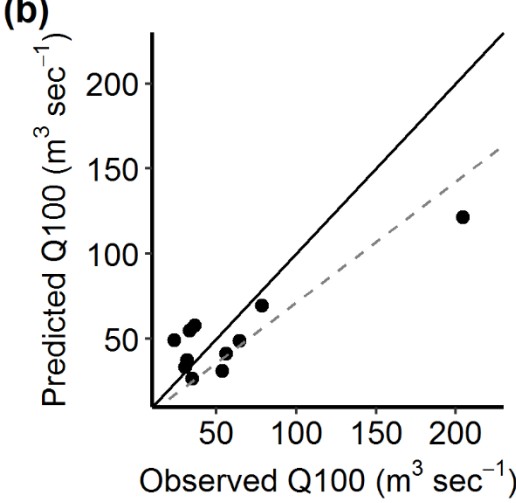


**Figure 2** – Observed versus predicted estimates for (a) Q2, and (b) Q100. The dashed grey line
depicts the linear regression between observed and predicted flow values, and the black, solid
line shows a 1:1 relationship.

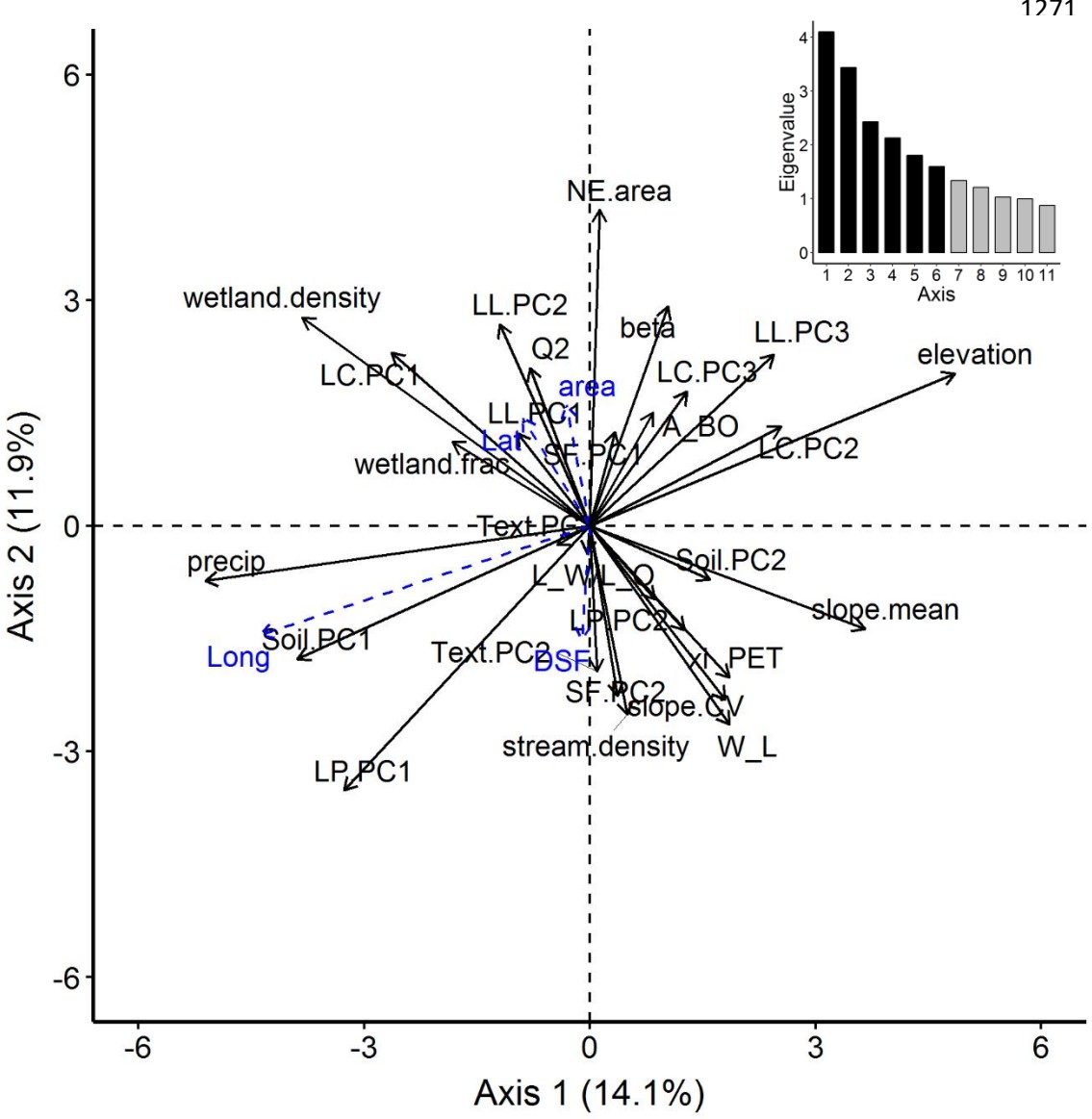

**Figure 3** – Principal components analysis for candidate variables for classification. Active and
supplementary variables are shown as solid black, and dashed blue arrows, respectively.
Eigenvalues for PC axes are provided (inset), with black bars denoting the six PCs used in the
hierarchical clustering analysis.


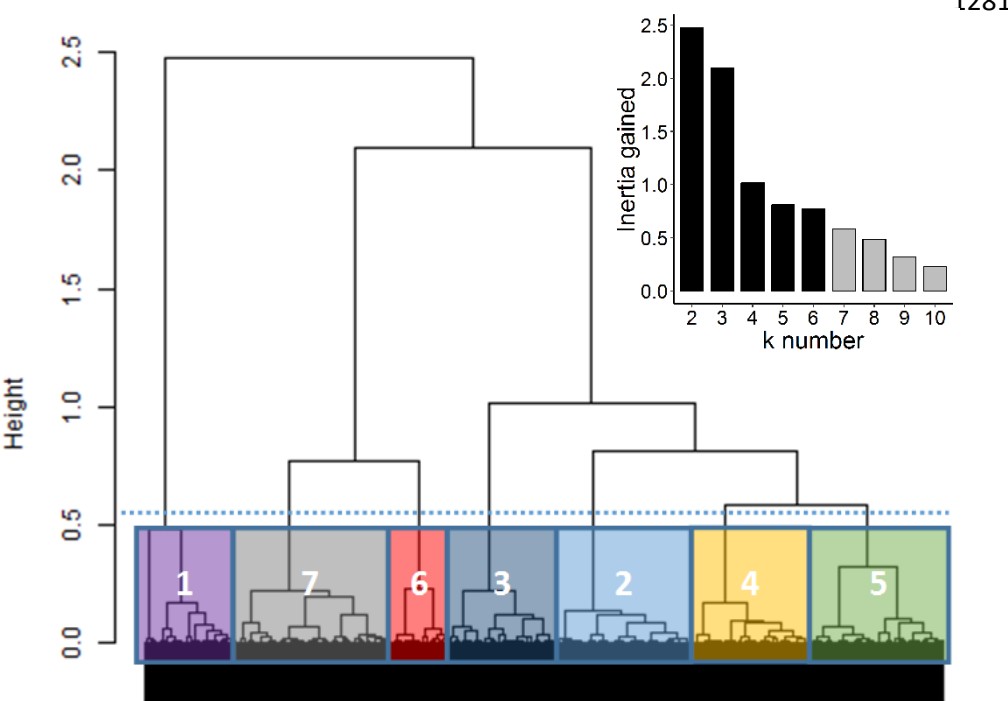

Figure 4 – Dendrogram resulting from the hierarchical cluster analysis of principal components. The blue, dashed line indicates the cut in the tree, resulting in seven clusters. The amount of inertia gained by increasing the number of clusters ($k$) is depicted in the inset panel.



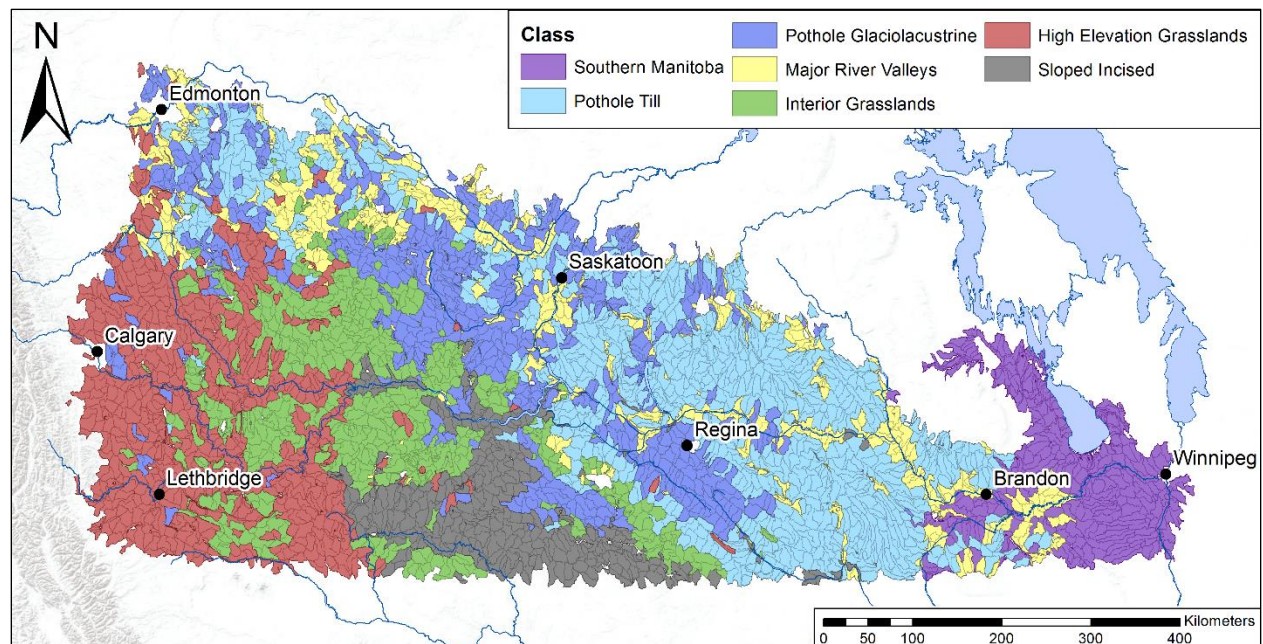

**Figure 5** – Classification of Prairie ecozone watersheds. Watershed delineations are from Lehner
and Grills (2013), available at www.hydrosheds.org. See text for detailed interpretation of the
seven clusters.

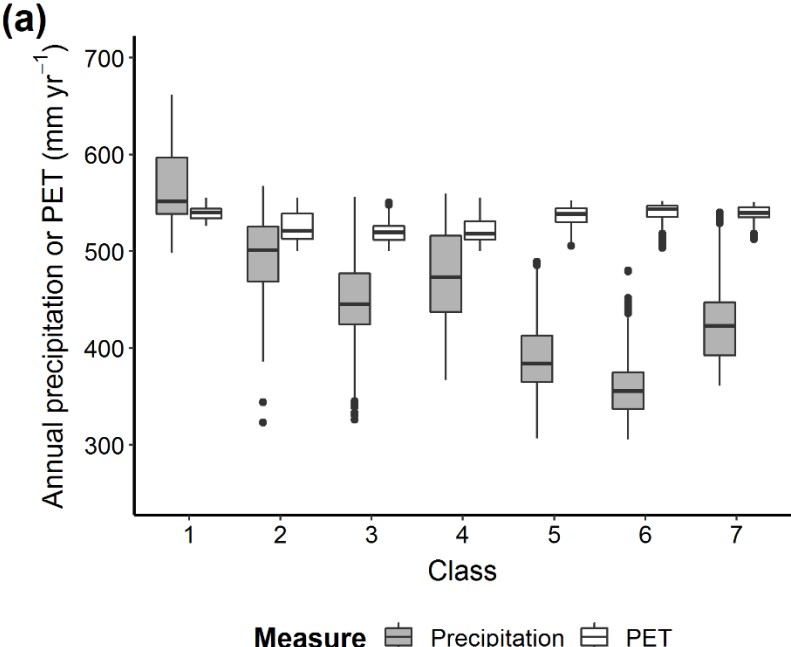

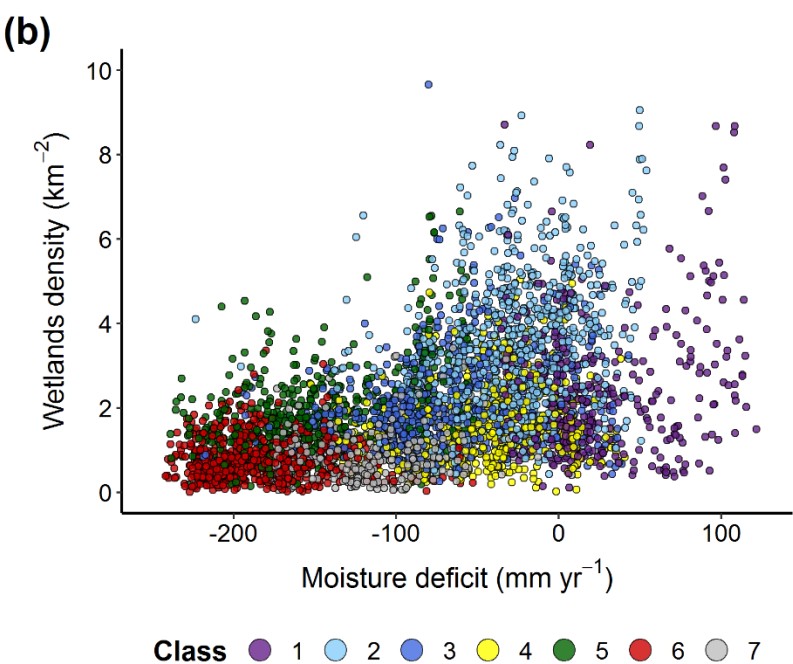


**Figure 6** – Climatic variation among watershed classes. (a) Boxplots of total annual precipitation
(grey) and potential evapotranspiration (PET) (white) for each watershed cluster. Lower, middle,
and upper limits of boxes show the 25th, 50th, and 75th quantiles, respectively. (b) Wetland
density to moisture deficit (Precipitation – PET). *Classes: Southern Manitoba (1), Pothole Till*
*(2), Pothole Glaciolacustrine (3), Major River Valleys (4), Interior Grasslands (5), High*
*Elevation Grasslands (6), Sloped Incised (7).*

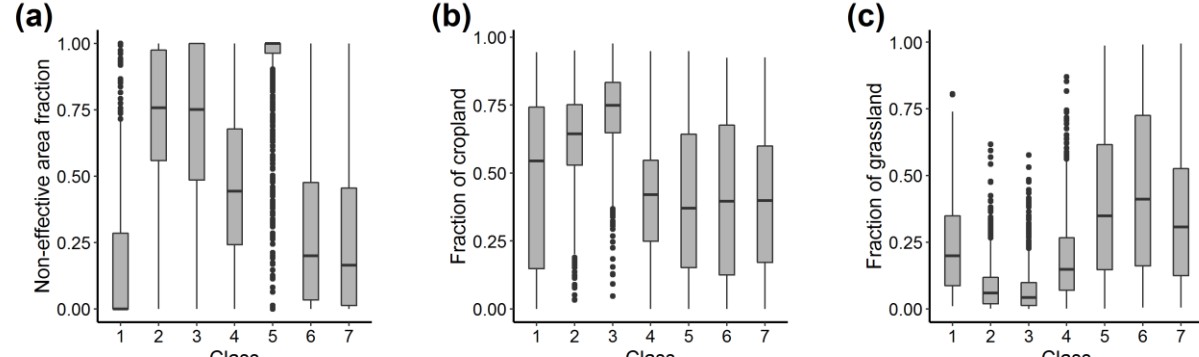


**Figure 7** – Boxplots of select variables by watershed class: (a) fraction of non-effective area; (b)
fraction of cropland; and (c) fraction of grassland. *Classes: (1) Southern Manitoba, (2) Pothole*
*Till, (3) Pothole Glaciolacustrine, (4) Major River Valleys, (5) Interior Grassland, (6) High*
*Elevation Grasslands, and (7) Sloped Incised.*

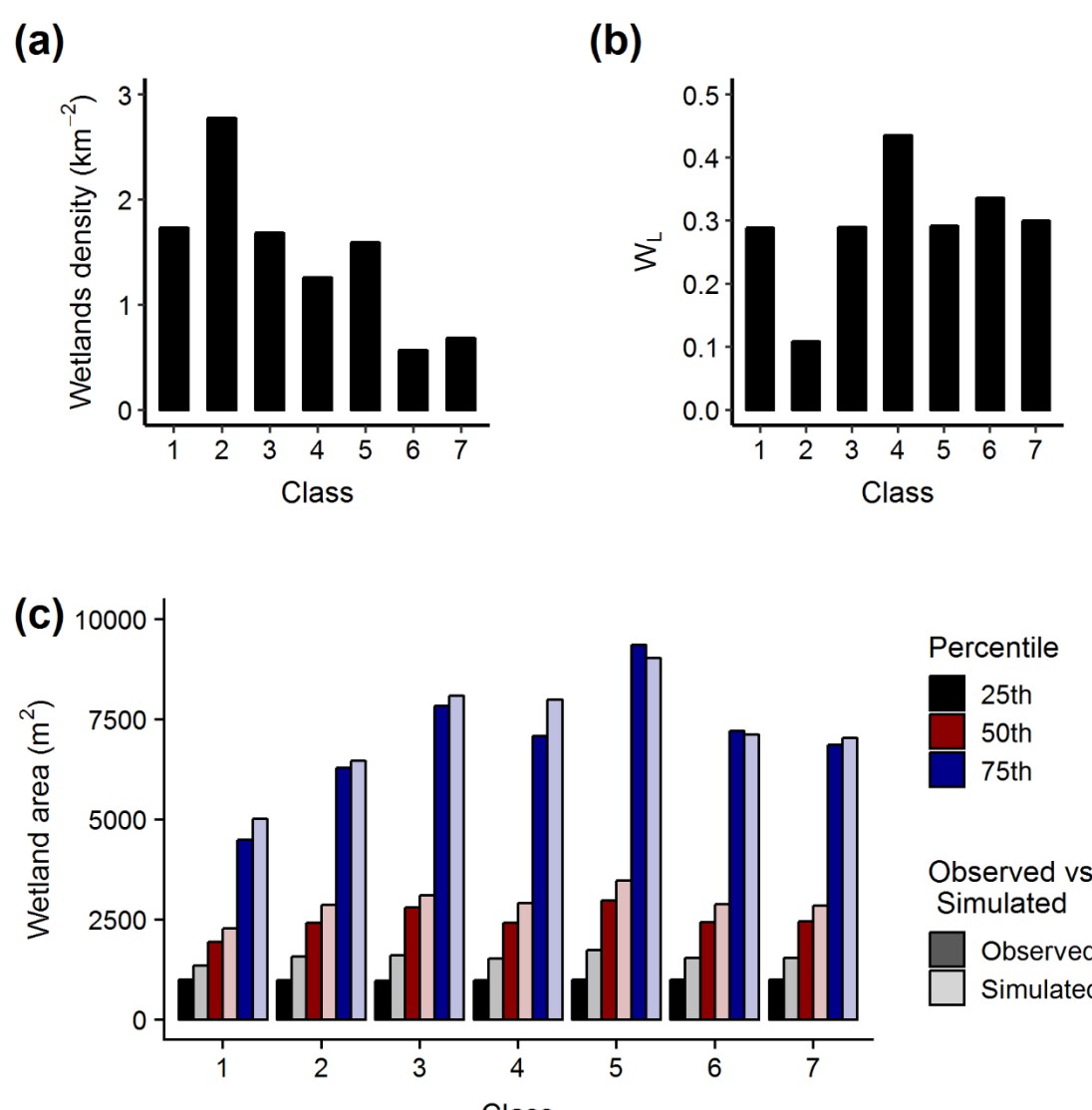


**Figure 8** – Wetland variables and simulated size distributions. Median (a) density of wetlands
and (b) fraction of total watershed water area in the largest wetland ($W_L$) are depicted by class.
Panel (c) shows observed (dark) and simulated (light) percentiles of wetland areas. Predicted
values are based on a generalized Pareto distribution and using median parameters of $\beta$ and $\zeta$ for
each cluster. Simulated data were restricted to the raster pixel resolution of observed data from
the Global Surface Water dataset. *Classes: Southern Manitoba (1), Pothole Till (2), Pothole*
*Glaciolacustrine (3), Major River Valleys (4), Interior Grasslands (5), High Elevation*
*Grasslands (6), Sloped Incised (7).*

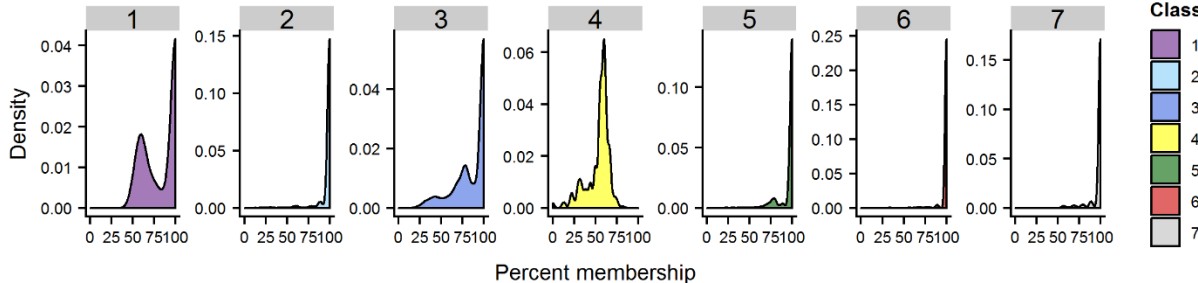


**Figure 9** – Density distributions of percent agreement of watersheds to the classification in Fig.
5 by watershed class. *Classes: Southern Manitoba (1), Pothole Till (2), Pothole Glaciolacustrine*
*(3), Major River Valleys (4), Interior Grasslands (5), High Elevation Grasslands (6), Sloped*
*Incised (7).*



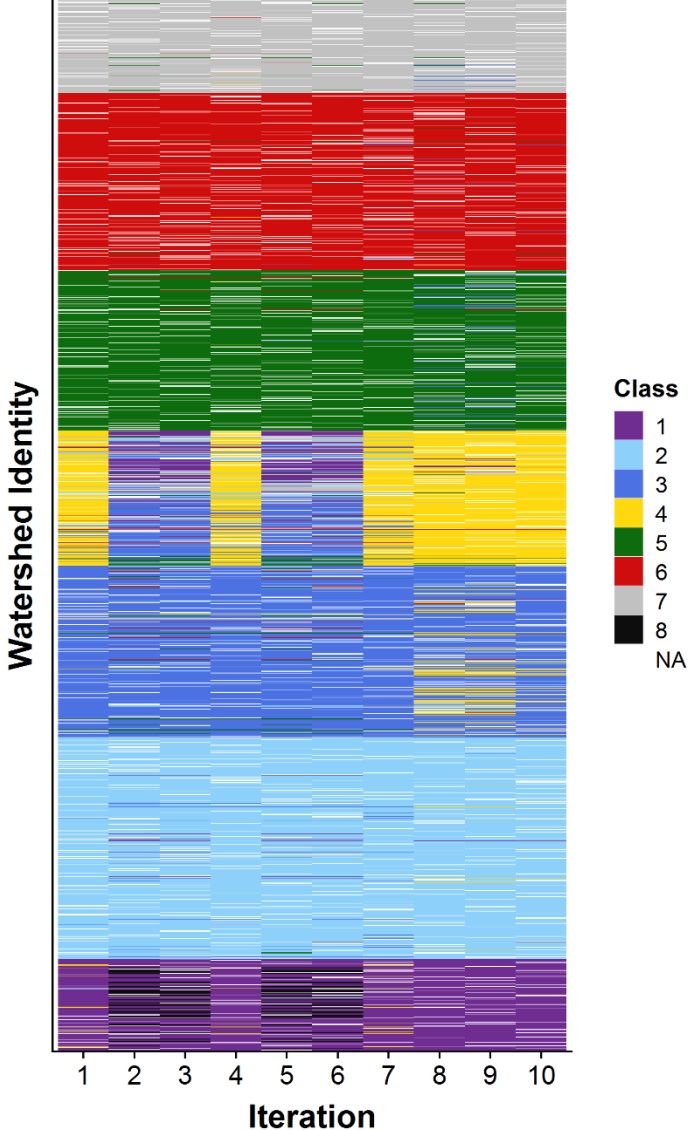


**Figure 10 –** Agreement of assigned watershed classification from the (original) complete
analysis, with class assignments from the iterative approach using re-sampling. Classes are
coloured according to that shown in Fig. 5, with those identified under a new class (C8) depicted
in black. Watersheds that were removed from the subsets analyzed are in white. *Classes:*
*Southern Manitoba (1), Pothole Till (2), Pothole Glaciolacustrine (3), Major River Valleys (4),*
*Interior Grasslands (5), High Elevation Grasslands (6), Sloped Incised (7).*