# Peer review of "A WATERSHED CLASSIFICATION APPROACH THAT LOOKS BEYOND HYDROLOGY: APPLICATION TO A SEMI-ARID, AGRICULTURAL REGION IN"

_Hydrology and Earth System Sciences, 2018_

## Referee Comment (RC1) · Hayashi (Referee) · 20 Jan 2019

GENERAL COMMENTS

This study aims at classifying the Canadian Prairie region into seven categories based on the information on climate, surficial geology, landform, soil type and texture, land use, wetland occurrence, and other factors. It is a land classification exercise, but presented as watershed classification because watersheds are used as elementary polygons. The study applies a suite of advanced statistical techniques to an array of spatial information data to come up with the classification that is consistent with the current understanding of the ecological and hydrological conditions of the Canadian Prairie. This is an impressive achievement resulting from a tremendous amount of time and efforts, and I can see the attractiveness of such approach. However, I am not sure what new scientific knowledge and insights we can gain from this study. The

authors try to demonstrate the usefulness of the new classification system in water management practices in the discussion section, but most of the findings have been qualitatively known to the community for a long time. Then, the usefulness must be in how the new classification can contribute to improving quantitative understanding or predictive capability, which is not explicitly demonstrated. It will be useful to examine the validity of the new approach using the 11 study watersheds, which I assume have hydrological data set. The data from the study watersheds are only used in Figure 8, and seem to be under-utilized. Aside from the issue above, I feel that the manuscript could use some clarification in methodology. Please see my specific comments below.

SPECIFIC COMMENTS

Line 102, 108. How is "watershed" defined? Is it straight forward to define watersheds in an unambiguous manner? Please clarify that here, or in the methods.

Line 117. How is the Canadian Prairie defined? Please present a brief definition, and the source of the ecozone boundary shown in Figure 1.

Line 119. The upper bound of precipitation (650 mm) seems to be too high. For example, mean annual precipitation in Winnipeg (wettest part of the Prairie) is only 601 mm for 1970-2000 according to the quality-controlled dataset (Mekis and Vincent, 2011. Atmosphere-Ocean, 49: 163-177). Similarly, the range of mean annual air temperature (5.7-7.4 C) seems too high. Based on the quality-controlled dataset, mean temperature for 1970-2000 is 2.9 C in Winnipeg, 2.5 C in Saskatoon, and 4.3 C in Calgary. It is well known that CANGRID dataset (ECCC 2017) has substantial errors and biases. I strongly recommend that the authors check the validity of CANGRID dataset at a few reference locations and discuss the potential implication of data bias.

Line 128. Related to my comments on Line 102 and 108, how are these watershed outlet selected? Please explain.

Line 136-138. As it is written, the sentence indicates that the watershed of the

Saskatchewan River is excluded from the analysis, which is clearly not the case. Please re-write the sentence to indicate more clearly what is excluded.

Line 140. Please indicate roughly how many kilometers are equivalent to 15 arc-second in the Canadian Prairie.

Line 141. The authors describe watersheds by referring the reader to Figure 1. However, Figure 1 does not show watersheds. Please refer the reader to Figure 5 instead, or add watershed boundaries to Figure 1.

Line 145. What is the total area of 4175 watersheds? How does that compare to the total area of the Canadian Prairie?

Line 156. Please see my comments above on CANGRID.

Line 161. Temperature-index methods such as Thornthwaite do not give reliable estimates of "potential evapotranspiration" compared to more physically based methods such as Penman-Monteith. There is nothing wrong with using the Thornthwaite method as a temperature index, but please explicitly acknowledge its limitation.

Line 162. The balance between precipitation and evapotranspiration is reflected in ecoregions of the Prairie, as plants are good indicator of long-term water balance. Ecoregions have been used by Prairie ecologists and hydrologists as a process-based framework for understanding the eco-hydrology of the region for many decades. It would have been very useful to use ecoregions as an element of classification, but the authors have chosen not to use them. Please provide an explanation.

Line 167. How were these non-effective areas determined? Please briefly explain the method and cite a reference. This is well known to Canadian Prairie hydrologists, but HESS is an international journal.

Line 177. Please change the wording to "seasonally flooded prairie potholes". Potholes are permanent landscape features, whereas flooded areas can be seasonal.

Line 180. Is (wetland density) needed here?

Line 191. Please briefly explain the meaning of mu and beta, and indicate the dimension or unit. These must have a unit of area to maintain the dimensional homogeneity.

Line 195. Is it true that all pixels in the Canadian Prairie have "monthly" satellite images? I do not think that is the case. Please clarify that in the texts.

Line 197. What do you mean by "the median area of the largest wetland"? Please re-phrase so the reader can understand what you mean.

Line 205. Surficial geology is mapped by geologists in each province using different terminologies. I am not sure if the "comparison across provincial boundaries" is straight forward. Please add a brief explanation on how the difference in terminology and mapping methods was reconciled.

Line 208. In the Canadian System of Soil Classification, colour indicates more than just an appearance of soil. For example, Black Chernozem and Dark Brown Chernozem are distinct soil types developed under distinctively different climatic conditions. The distribution of these soil types often coincides with ecoregions (e.g. Black Chernozem is associated with Aspen Parkland). Please consult with local soil scientist to give a better context to soil classes. Also, somewhere in the paper, perhaps near the beginning of the method section, it will be useful to present a process-based framework to understand the eco-hydrological functions of the Canadian Prairie landscape (see my comment on Line 162).

Line 223. Please indicate the unit of DSF. It must be the inverse of length.

Line 255. Please indicate these prairie stations in Figure 5. I assume these are the "study watersheds" described in Line 472. Please point that out here.

Line 265. Please explain how V1 and V2, and W1 and W2 are defined. Please note that most readers of HESS are not familiar with CCA. You do not have to present detailed explanation of CCA, but you need to give a brief outline so that the reader can

understand the basic concept.

Line 266. What are "the original variables"? Please explain, using a table if appropriate.

Line 290. ". . . attributes and is the basis . . ." for matching the tense.

Line 301. Please define alpha.

Line 310. What does this mean? Based on Line 269, does it mean that the result was very useful for V1-W1, and barely useful for V2-W2? Please explain.

Line 311. What correlation value would indicate "strong"? Does it have a statistical level of significance, like in the standard correlation analysis? Does a negative value indicate negative correlation? Please explain.

Line 311-312. It is true that the correlation value is strong between Q100 (1:100 flow) and W2, but it is weak for Q2 (mean annual flow) and W2. On the other hand Q2 and W1 has a strong correlation. Also the lambda value is much greater for V1-W1 combination than for V2-W2 combination. Given that, why was W2 chosen? Is it because the classification is designed for 1:100 flood prediction? Please provide an explanation.

Line 322. How is rock fraction area calculated? I cannot imagine there are many areas of exposed bedrock in the Canadian Prairie. Please explain.

Line 326. Please list the classes of surficial geology used in the analysis.

Line 347. What are the "PCs from compositional datasets"? Are these different from PC1-PC6 in the header of Table 3? Please explain.

Line 358. "Weaker", not "less strong".

Line 389. The Canadian Prairie has now been divided into seven classes, which seem to be consistent with our current understanding of eco-hydrology. For example, C1 roughly coincides with the ecoregion "Lake Manitoba Plain (162)" in the Ecozones and

Ecoregions of Canada (Ecological Stratification Working Group, 1995). Then, what new knowledge and insights can we learn from this exercise? It will be nice to see a clear demonstration of the contribution of this study to new advances in "Hydrology and Earth System Sciences". Please try to present that in the discussion section.

Line 412. Glacial till and hummocky landforms. Does this refer to one thing, or two separate things (till and hummocky landforms)? Hummocky landform is a sub-class of glacial till terrain. Please clarify.

Line 453. Brown Chernozem is associated with the "Mixed Grass (159)" ecoregion, which covers much of the driest part of the Canadian Prairies, commonly referred to as the "Palliser Triangle". Accordingly the outer boundary of C5 roughly coincides with the outer boundary of Mixed Grass. However, Figure 5 shows a patch of C6 in the core of the Mixed Grass, which is the driest part of Alberta having distinctly different eco-hydrological characteristics compared to the band of C6 parallel to the western boundary of the Prairie. Is the new method picking up new information, or is it erroneously classifying watersheds? Are there too many classes in the system? These are worth discussing in this section.

Line 472. Are there 11 study watersheds, as indicated in Line 255? If so, is that a high enough number to examine all seven classes? Please explain.

Line 490-493. It is true that few studies have classified "watersheds" in the prairies, but there have been numerous studies examining the spatial distribution of eco-hydrological functions of the Prairie landscape. For example, ecoregions are an integral measure of hydro-climatology. Please acknowledge previous efforts and highlight the newness of this work.

Line 502. This is an example demonstrating the strong effect of ecoregions on hydrology.

Line 633. Yes, but the delineation has been available for many decades in the form of

ecoregions. Please acknowledge it.

Line 637. Geography may not be an appropriate term here, because geography encompasses many things, not just landforms. I would say topography or landform is more appropriate.

Line 661. Figure 8 just shows wetland density and area delineated in satellite images, which is dependent of climatic factor (wetness) in addition to depressional storage capacity. Overall, I believe that the data from the 11 study watersheds can be utilized more to demonstrate the validity and usefulness of the new classification method. For example, are there distinct differences in the hydrological characteristics of seven classes of watersheds?

Reviewer: Masaki Hayashi

―――――――――――――――――

---

## Referee Comment (RC2) · Knoben (Referee) · 21 Jan 2019

General comments

1. This study uses clustering analysis to define regions of hydrologic similarity in the Canadian Prairies. The authors distinguish seven classes, using information on the region's climate, wetland traits, topography, land use, and streamflow. The authors use a range of techniques to reduce dimensionality of the data and create an attractive map of hydrologic/ecologic similarity in the Prairies. I appreciate the enormous effort that must have gone into this work, but I'm also left with several concerns that will need to be addressed, focusing on the validity of the clusters and clarification of the methods.

2. Discussion of the data preparation and initial analysis is extensive, whereas the assessment of how well the clustering has worked is comparatively small. For any practical application of their results, a measure of the validity and usefulness of these

clusters is needed, beyond the discussion given in section 3.5 (which mainly lists which attributes were most informative in creating the clusters). A quantification of usefulness through independent data, or an example application of these clusters as a management tool would improve this manuscript by a large margin. I have outlined a possible option in the specific comments.

3. The methods section is also partly used to introduce data, and to define the metrics used for classification. Readability of the manuscript might be improved if these two are separated (i.e. a data section followed by methods) or if the authors provide a table that lists their classification metrics, the data the metrics are based on, and the hydrologic relevance of each metric. Currently, it is not entirely obvious to me why certain metrics are included in the analysis and why others, that seem obvious candidates to me, are not (detailed in the specific comments). I assume this is obvious to experts on prairie hydrology, but a little more background would make the manuscript accessible to a wider audience.

4. The authors present a lot of information in section 2, and not all readers will be equally familiar with all concepts. A bit more background might help those without extensive knowledge about the specific methods used in this work. Additionally, the results section could benefit from a brief introduction (or figure) that outlines what is coming. E.g. "First, in section 3.1 we use CCA to . . . Next, we reduce the dimensionality of our problem through PCA (section 3.2). . . .".

Specific comments

1. International readers might not be able to place the Canadian Prairie on a map (line 55). A brief statement about the geographical extent of the Prairies would help.

2. "Hydrological characteristics" (line 71) is unclear. Do the authors mean catchment attributes (e.g. topography, soils), climatic conditions, statistical properties of the streamflow regime or something else?

3. It would be helpful for the reader to briefly summarize how well earlier classification attempts have worked (line 74-78) and where the authors see current challenges.

4. The HydroSHEDS webpage (https://www.hydrosheds.org/page/development) lists a few regions where the data set is prone to errors, including areas with low or not well-defined relief. Is this of concern in the Canadian Prairies?

5. Approximately how many meters are 15 arc-seconds (line 140) in this area?

6. What motivated the choice for these specific area (line 142) and urbanization (line 143, Table S1) thresholds?

7. The spatial resolution of climate data (line 157) seems large compared to the resolution of the watershed boundaries. Can climate data on this resolution still be considered representative for the smaller catchments?

8. What is the rationale for choosing the Thornthwaite method (line 161)?

9. Snow formation and melt can strongly influence the seasonal water distribution and accounting for the fraction precipitation that occurs as snowfall has recently proved valuable in hydrologic similarity research (Knoben et al, WRR, 2018; https://doi.org/10.1029/2018WR022913). Is there any particular reason why the authors use only mean P and ET in their clustering?

10. What is meant with a wet cycle (line 176-177)?

11. Please include a (short) definition of potholes (line 177).

12. Why is the Lw/Lo metric (line 184) relevant? What does this metric tell us about watershed behaviour?

13. The climate data (line 156), land cover data (line 230 and further) and hydrological data (line 252 and further) cover different periods in time (1970-2000 for climate, 2011/2016 for agriculture land use, 1990-2014 for hydrologic data). For a general classification of similar regions, overlapping time periods for the data sources would be

more appropriate. What is the rationale for not doing this?

14. Estimation of mean flow Q2 and flood Q100 (line 252) for 4175 watersheds using only 11 stations (line 255) seems ambitious to me. Spence and Saso (2005) show a significant uncertainty in their predictions. Can the authors provide a statement about their confidence in the Q2 and Q100 estimates?

15. What is the reasoning behind the 80% threshold for PCA components (line 279)? Perhaps the authors can include a plot or table that shows the importance of each PC to support this choice.

16. Were variables standardized to a fixed interval (e.g. [0,1]) in addition to the log-transform (line 282)?

17. Line 286-287 needs clarification. Which variables are the "complete suite of variables"? The previous section gives the impression that all variables were converted to PCs, of which only those above 80% would be used. A table with a summary of all variables used, their data source(s) and their hydrologic relevance could help clarify what is going on.

18. Retaining PCs above 50% (line 291) seems to contradict retaining PCs above 80% (line 279).

19. A short description of Ward's criterion (line 295) would be helpful.

20. I suggest replacing "and thus did not explicitly affect the clustering analysis" (line 303) with "and are not included in the clustering procedure" (assuming that I correctly interpreted this sentence).

21. Not all readers will be equally familiar with canonical regression analysis. I find it difficult to interpret the results in section 3.1. A (very) brief description of CCA might help. Some questions I'm stuck with: are those lambda values high or low? What would either tell us? What does it mean that hydrologic variables are associated with W2?

22. I would say these regressions are not particularly convincing (line 314). It looks as if the one high value could be inflating the correlation value. Did the authors use Pearson or Spearman correlations? Predicting streamflow characteristics in ungauged basins (i.e. regionalization) is an active field of study but achieving robust results has proven very difficult. How does this impact the extrapolation of this information to the 4100+ watersheds and what are the consequences for the subsequent analysis?

23. Section 3.2 (PCA results) lacks a logical conclusion (or perhaps an introduction). How did the authors choose how many PCAs to discuss and which PCAs are selected to be used in subsequent steps?

24. The difference between active and supplementary variables needs to be defined (line 348).

25. Section 3.3 lacks a logical conclusion. Which PCAs are carried over to the clustering analysis?

26. What do the authors mean with "definition of clusters" (line 370)?

27. Section 3.4 is very brief. One of the main aspects of clustering analysis is assessment of how good the resulting clusters are. Currently the authors extensively list the differences between the clusters (section 3.5) by summarising which inputs were most influential in determining the clusters. However, this only tells us something about the patterns in the data and not much about the usefulness of these clusters. The authors suggest in the discussion that these clusters can be helpful to inform management decisions, by showing which regions are expected to behave similarly and which regions are not. This statement should be backed up by proof with independent data that these cluster indeed show that. The GSIM archive (Do et al, HESSD, 2018; https://doi.org/10.5194/essd-10-765-2018) is a recent contribution of global streamflow indices which might provide the authors with independent hydrologic information that they can use to quantify how well their clusters group hydrologically similar regions. See e.g. Knoben et al, WRR, 2018 (linked above) for possible ideas.

28. The subsections of section 3.5 are hard work for an international audience. Perhaps figure 5 can be expanded to include a map which shows the various names used in these sections (see e.g. Addor et al, HESS, 2017; figure 1e; https://doi.org/10.5194/hess-21-5293-2017 )

29. Line 435-437 ("Being river valleys … Q2 values (Tabl 1)) repeats line 428-429.

30. I'm unsure how section 3.6 relates to the previous clustering results. I was under the impression that wetland density is one of the variables used during clustering. Should section 3.6 perhaps be moved to before the clustering results? Also, if this is part of the clustering analysis (as e.g. table 3 and 4 seem to suggest), why does this specific attribute deserve its own section? Edit: reading back, it seems to me that wetland distributions were estimated (line 186 and further). In that case, are the observations referred to in line 480 from the 11 stations? This seems a small sample of observations to compare results for 4100+ watersheds to. How confident can we be in these estimates?

31. The authors stress the importance of accounting for human influences (Section 4.1) in classification procedures. Can they comment on the extent to which this was done in their work and do they have any recommendations for future efforts? For example, should artificial drainage density be considered as a variable?

32. The authors mention that certain variables can dominate the clustering approach (line 579 and further). This is why it is not uncommon to standardize clustering variables to a fixed interval, because this reduces the effect of a variable's variability. Log-transforms lessen, but do not prevent this. Can the authors comment on which variables had the widest (log-transformed) range and whether this correlates with the variables that are most important during clustering?

Technical corrections

1. "van der Kamp" (line 48 and others) should be "Van der Kamp ".

2. Figure S1, c: text in the centre overlaps and is unreadable.

3. "described" (line 342) should be "describe".

4. Figure 6b: the number of points make this plot difficult to read. x-axis should be changed to cover the width of the page. Possibly cut of the y-axis at 10 for additional clarity.

5. "Bering" (line 435) should be "being".

6. Figure 7. Readability would be improved if the numbering of classes is placed in front of the class name (like was done for subplots a, b, c).

7. Figure 8. Suggest changing "solid" to "dark" and "transparent" to "light".

8. Figure 8c. A boxplot seems more appropriate than these bar plots given the information presented.

9. Line 540. Remove "a".

10. Line 603. Change to "... may inadvertently result in the presence of smaller wetlands being perceived as ..."

11. Line 662-663. Some words need to be removed to make the sentence make sense.

12. Line 696-697. This sentence needs rewriting.

---

## Referee Comment (RC3) · Anonymous Referee #3 · 15 Feb 2019

This study focuses on the classification of watersheds into homogeneous regions sharing the same climatic and physiographic characteristics. While well-structured and well written, this paper does not add very much to the existing knowledge. Moreover, the proposed approach has some fundamental issues that need to be vigorously addressed: 1) Ambiguity: It has been mentioned that the CCA was used for estimating hydrologic variables since only a few observing stations are available. These variables will be considered later in the classification approach to provide a watershed classification system that will be used, among other purposes, to estimate the hydrological response of a given watershed. What is confusing and contradicting here is to first estimating hydrological variables, and then using classification outputs to understand the hydrological behavior! A regionalization approach is more suited for this purpose. 2) I feel inconsistency in using CCA (the most appropriate classification method as recognized in regionalization studies) to estimate hydrological variables, and using another

classification method, hierarchical cluster analysis, for classification. 3) Equation in Line 319 is not very convincing since no precipitation or water-related variable is introduced. Also, only 11 observations have been considered for calibration. Assessment of the uncertainty is not consistent too.
* * *

---

## Author Comment (AC1) · 26 Mar 2019

General response

We would like to thank the three reviewers for their insights on our manuscript. Many of the suggestions were constructive and will contribute towards building a stronger study. A few comments were shared among the reviewers. These shared comments were: (1) utilizing observed data more effectively to evaluate potential differences in class hydrological response, (2) elaborating on the new knowledge gained from the current study, and in particular the relevance to an international audience, and (3) increase detail on the methods and evaluation of the CCA. We summarize how we will address these common comments here, and provide additional responses to each individual reviewer in the sections that follow.

Specifically, we will consider the appropriate use of available independent data to evaluate the separation among our clusters, and potentially resampling approaches to assess class membership. We must emphasize, however, that previous attempts to establish, or validate classes based on measures of streamflow data suffer real limitations due to the hydrological setting with the Prairies, which features a wetland-dominated landscape and spill-fill dynamics. Alternatives to streamflow data are sparse or non-existent, and this scarcity is central to the motivation of our study. We seek to differentiate among watersheds within the Prairie and to do so in a way that connects to decision-making at local scales, as explained in our Introduction. However, with recognition of these limitations, we believe that independent assessment of the classes (Reviewer 2) will strengthen our approach. Expanding on Figure 8 is one way to address the concern. We will re-evaluate how we can use observed hydrometric data and the across class patterns.

Regarding international relevance, we can place more emphasis on the use of our classification for serving as a basis more parametrizing hydrological models that provide insight into hydrological response on a more localized (100km2) scale than what is given by ecoregions (see response to Referee #1). Highlighting how our approach addresses these concerns will also emphasize "new knowledge" provided by our work to an international audience, especially to arid regions that might face similar challenges where stream networks are not well-developed.

Technical questions related to the Canonical Correlation Analysis are outlined in our responses to Referee #1 and #3. We will clarify the use of the "study watersheds" and associated observed data to the CCA. We can also provide justification for the inclusion of this procedure in our analysis, in turn providing added detail to our methods.

Response to Referee #1 - Summary

We thank Referee #1 for their helpful comments on our manuscript. We respond to the suggestions individually below and where appropriate, we indicate how we intend to enhance clarification in the text. In the summary, we address three key comments

from Referee #1: (1) reflecting ecoregions in the Prairies as units for eco-hydrological response, (2) clarification on datasets used, and (3) utilization of the 11 watershed for comparison to classes.

Referee #1 drew attention to the past use of ecoregions as units to describe regional eco-hydrology. In particular, they ask how the present study offers new insight. Our analysis provides a quantitative approach that evaluates physio-geographical variables of watersheds and the organization of smaller scale watersheds into broader arrangements (i.e., classes). Although some class boundaries might align with those of ecoregions, others do not. In addition, we sought to identify watersheds that exhibit similar characteristics, or classes, that are transferable across the Prairies. Here, we summarize 7 classes based on watershed areas of ∼100km2, which is a finer scale than that offered by ecoregions. We also include additional information that are not incorporated in ecoregion classification, such as wetland size distribution parameters and tillage practices, which directly affect hydrological dynamics of Prairie watersheds. However, we recognize comparing ecoregions to our findings, particularly the spatial extent, can provide geographical context for international audiences, as well as offer the opportunity to discuss our approach in the context of ecoregions, which are used across disciplines. We intend to amend the Discussion to include pertinent references, as well as perhaps provide a visual representation of local ecoregions (e.g., additional figure or amend Figure 1). We can also emphasize the use of our classification as serving as a basis more parametrizing hydrological models that provide insight into hydrological response on a more localized (100km2) scale, which is more relevant to managers working at this spatial extent.

Referee #1 asked for added clarification in our descriptions of surficial geology (Line 205) and soil zone (Line 208) datasets. We appreciate the reviewer's insights and believe added detail here will strengthen our manuscript. Amelioration among surficial geology definitions was performed by grouping more defined classification into broader categories describing depositional features. Grouping was performed by comparing definition of each feature type using the provincial government metadata and informed by advice from a colleague in geology. We acknowledge that these are broad groupings and working with a framework used coherently across the provinces would be ideal. However, for our current purposes, these broad descriptions were useful in capturing a variation across at least broad geological settings. In addition, it is true that hummocky landforms are associated with glacial till deposits (Line 412). However, the landforms dataset describes forms that include aspects of surficial geology, relief, among others. Therefore the two datasets are related. We feel that both datasets offer information on local geography. The hummocky landform designation is particularly useful for characterizing landscape influences on depressional storage and overland flow.

In regards to soil zones, we recognize that the "colour" is only a descriptor and the function of the soils are different among soils types. Importantly, soils develop under specific climatic conditions, geology, and vegetation, and these considerations are implicit in the data that we used. We will add clarification to the sentence, and elsewhere as appropriate, to elaborate that these data extend beyond just "colour". We also note that soil texture class was also used to describe soil characteristics.

We appreciate the suggestion for utilizing observed data more effectively within our study. In particular, Referee #1 references the "11 study watershed". We note the "study watersheds" in Line 473 is misleading. Here, we are referring collectively to the 4100+ watersheds used in the clustering analysis. However, the 11 streamflow stations were only used in the CCA. We will edit the sentence for clarity, and we will consider how we might incorporate observed data to compare the identified classes. An issue using observed data is that the reference watersheds do not necessarily compare to our scale, and the premise of the approach of our study is based on the limitations of previous approaches. The 11 stations are those with good quality streamflow data, while they are also at a scale that is not completely representative of prairie watersheds. In particular, the stations to not represent those watersheds not connected to

well-drained, stream-dominated systems, such as Interior Grasslands class. In these cases, the 11 stations would not be an appropriate comparison. We can therefore try to seek alternatives to providing an example of the hydrological behaviour or differences among classes. We can utilize more remotely-sensed wetland data (as shown in Fig. 8) or jack-knife approaches to evaluating watershed class membership. We will evaluate the benefits of potential alternatives as we revise our manuscript and consider the three reviewer comments.

Finally, we appreciate the technical suggestions given by Referee #1. We will incorporate these edits into our manuscript.

---

## Author Comment (AC2) · 26 Mar 2019

Response to Referee #2 - Summary

We greatly appreciate Referee #2's comments and feedback on our study. Please see our response to Referee #1 for suggestions shared among reviewers. Referee #2 provided a number of useful insights on the classification procedures and suggestions to appeal to international audiences. In particular, defining of terms related to prairie hydrology was suggested under "Specific Comments" (e.g., 1, 11, and 12) and we will take these into account as we revise the manuscript. We also appreciate the reference to Addor et al. (Comment 28) as an example of a classification study with an international context. We will consider appropriate strategies to make section 3.5 more applicable to international audiences, such as a map of ecoregion, and locations named in the section.

[Figure]

Referee #2's insights into clustering approaches and classifying watersheds is very valuable. Comments related to the clarification in methods can be improved on by adding details into this sections as per the suggestion by reviewers. The referee suggested that readability of the manuscript might be improved if the data sources and methods are separated into two sections. Readability might also be enhanced via a figure/diagram to show the workflow of the classification procedures. We agree that these suggestions offer improvement and will re-structure accordingly. Although a full explanation of the CCA method is beyond the scope of the paper, we will reference key literature or studies for readers interested in a more detailed description.

A useful suggestion from Referee #2 was to relate classes to patterns in observed data to evaluate usefulness of the method. We appreciate the reference to the GSIM archive (Do et al. HESSD), which offers useful procedures for comparing classification methods. We have access to the HYDAT dataset, which was used in the Do et al. study for Canadian data. We will consider comparing data from stations that are relatively undisturbed and reliable time series. An issue with the Prairies is that many reference hydrological stations are confined to main river systems and not necessarily represented of the behaviour of wetland-dominated (Pothole Classes), or more arid classes with low effective areas (Interior Grasslands Class). This complication informs why we chose to focus on predicting depression size distribution (Fig. 8) with observed data. We will evaluate the applicability of some independent data sources, (e.g., HYDAT, wetland remote-sensed data) to compare our classes.

Another set of suggestions concerned added clarification on the use of climate dataset (e.g., 7-10, 13). In particular, we thank the reviewer for the suggestion to consider changing fraction of precipitation as snowmelt and reference to Knoben et al. 2018. We agree that inclusion of this parameter is and likely valuable for the Prairies. We focused solely on precipitation and ET because these variables were available at the temporal length and spatial extent for the study. Given the limitations of the dataset we used (as alluded to by Referee #1), calculating parameters at a seasonal scale

might introduce more uncertainty, and thus was not included here. However, fraction of snowfall should be considered in future iterations provided the data is available at a suitable spatial resolution.

Referee #2 provided a number of technical suggestions. Suggestions include discuss the limitations in the HydroSHED dataset (4), which would be a useful addition to our manuscript due to the limitation in SRTM data accuracy in the context of the Prairies. We will provide a discussion of this concern. They also identified incidences of regional terms, such as "pothole" and metrics like Lw/Lo, where the significance might not be obvious to those not familiar with the area. We will give attention to those terms, and provide definitions where necessary.

We will also address the other useful technical comments provided in order to strengthen the clarity of our manuscript.

---

## Author Comment (AC3) · 26 Mar 2019

Response to Referee #3 - Summary

Comments from Referee #3 focused on the Canonical Correlation Analysis (CCA). We appreciate their request for more detail regarding our analysis, and we discuss some shared concerns among reviewers in our response to Referee #1. Although a complete explanation of the CCA method is beyond the scope of the current work, please see below regarding our responses to Referee #3's feedback.

In order to reduce the ambiguity in how we applied canonical correlation analysis in this study we will rewrite the section describing the approach. Canonical correlation analysis was used for the purpose of estimating mean annual runoff and the 1:100 year flood for the 4175 watersheds because it was felt that it provided a more independent means of regionalization than using terms directly applied within the subsequent cluster

[Figure]

analysis. In regards to any inconsistency in using two methods - canonical correlation analysis and hierarchical cluster analysis - the former was used for the regionalization exercise to derive streamflow estimates for each watershed and the cluster analysis to classify the watersheds. The latter was not used to regionalize flows, and is considered a better tool for this purpose than canonical correlation analysis. Overall, we believe that addressing these concerns and providing added clarity in our methods will enhance our manuscript. We appreciate the feedback given by the reviewers.

---

## Author Response (AR1)

**Response to reviewers: "Watershed classification for the Canadian Prairies"**
*Please note that we have changed the manuscript title to: "A WATERSHED CLASSIFICATION*
*APPROACH THAT LOOKS BEYOND HYDROLOGY: APPLICATION TO A SEMI-ARID,*
*AGRICULTURAL REGION IN CANADA".*
**Response to Referee #1**
**Response to GENERAL COMMENTS**
*We thank the reviewer for their comments, and we appreciate the time taken to provide them. Yes, these*
*traits of the Canadian Prairie may have been known by select individuals qualitatively for some time, but it*
*is necessary to conduct this analysis quantitatively so as to begin to address some of the most pressing*
*water management issues on the Canadian Prairie. This manuscript alone is a sizeable body of work,*
*requiring careful and lengthy description. Extension to an application of the classification would render a*
*single manuscript unwieldy. Applied use of the classification results will be pursued in subsequent papers.*
*We agree that one of the scientific contributions of this work is in improving quantitative understanding of*
*classifications in this region, which is why we expanded discussion of comparisons to previous*
*classifications in this new version.*
**Response to SPECIFIC COMMENTS**
Line 102, 108. How is "watershed" defined? Is it straight forward to define watersheds
in an unambiguous manner? Please clarify that here, or in the methods.
*We thank the reviewer for their comments. We have added clarification on operative definition of*
*watershed used here in the methods, as well as additional detail on derivation of watershed*
*boundaries.*
Line 117. How is the Canadian Prairie defined? Please present a brief definition, and
the source of the ecozone boundary shown in Figure 1.
*We have added a brief description on the ecozone, including vegetation, to section 2.1. The*
*source for the ecozone boundary has been added to Figure 1.*
Line 119. The upper bound of precipitation (650 mm) seems to be too high….
*We have changed the value in the sentence and those of mean annual air temperature and*
*provide clear references to the source of these statistics.*
Line 128. Related to my comments on Line 102 and 108, how are these watershed
outlet selected? Please explain.
*We define the use of "outlet" for the purpose of this study on section 2.3.2., whereby it is the*
*lowest elevation along the watershed boundary.*
Line 136-138. As it is written, the sentence indicates that the watershed of the Saskatchewan River is
excluded from the analysis, which is clearly not the case.

*We thank the reviewer for this suggestion, and agree that the sentence was misleading. We have*
*removed the sentence and adjusted text for clarity.*
Line 140. Please indicate roughly how many kilometers are equivalent to 15 arcsecond
in the Canadian Prairie.
*We thank the reviewer for this comment, which was shared by Referee #2. We provided the*
*metre equivalents at Saskatoon, Saskatchewan, which is located within the Prairies ecozone. The*
*paragraph now reads: "Delineations of candidate study watersheds were obtained from the*
*HydroSHEDS global dataset (Lehner and Grill 2013). Watershed boundaries within this dataset*
*were based on Shuttle Radar Topographic Mission (SRTM) digital elevation model (DEM)*
*calculated at a 15 arc-second resolution. The resolution is equivalent to for example*
*approximately 285 m east-west and 464 m north-south at Saskatoon, SK."*
Line 141. The authors describe watersheds by referring the reader to Figure 1. However,
Figure 1 does not show watersheds. Please refer the reader to Figure 5 instead,
or add watershed boundaries to Figure 1.
*We have removed the reference to the figure at line 141 as it was decided to be unnecessary.*
Line 145. What is the total area of 4175 watersheds? How does that compare to the
total area of the Canadian Prairie?
*The area for the Prairie ecozone (4.7 x $10^5$ km$^2$) and the watersheds included in the study (4.2 x*
*$10^5$ km$^2$) are now provided.*
Line 156. Please see my comments above on CANGRID.
*CANGRID is the only gridded product data that uses the Adjusted Homogenized Canadian*
*Climate Dataset, and we felt it the most appropriate to use in this region where precipitation*
*undercatch in gauges is very pronounced. We have added clarification in the text.*
Line 161. Temperature-index methods such as Thornthwaite do not give reliable estimates
of "potential evapotranspiration" … please explicitly acknowledge its limitation.
*This acknowledgment was addressed by including the following sentences: "To maintain*
*consistency among climate data, and use the same temperature data as described above,*
*options were limited with which to calculate PET.  PET was calculated from the Thornthwaite*
*equation (Thornthwaite 1948) using the SPEI package (Vicente-Serrano et al., 2010). A*
*disadvantage of the Thornthwaite approach is it assumes a correlation between temperature and*
*radiative forcing and adjusts for any lag in this relationship using corrections for latitude and*
*month."*
Line 162. The balance between precipitation and evapotranspiration is reflected in
ecoregions of the Prairie, as plants are good indicator of long-term water balance.
… Please provide an explanation.
*Please see above for a more detailed explanation on ecoregions. Briefly, we acknowledge*
*vegetation as an indicators of the water balance. However, in the Prairies, much of the local*
*"natural" vegetation in not reflected due to human land modification (e.g., agriculture). We use the*
*landcover types from AAFC to consider portions of the natural vegetation, such as woodlands*
*and grasslands.*

Line 167. How were these non-effective areas determined? Please briefly explain the
method and cite a reference. This is well known to Canadian Prairie hydrologists, but
HESS is an international journal.
*These were defined by (Mowchenko and Meid, 1983). We will include this citation and provide a*
*brief description. We also provide more detail in Section 2.3.2 as to the impact of non-effective*
*areas to prairie hydrology, and we included the following description: "The location of these*
*regions are shown in Figure 1. This definition stems from work by Agriculture and Agri-Food*
*Canada where prairie drainage areas were divided into gross and effective drainage areas,*
*whereby the former describes the divide that is expected to contribute under highly wet condition,*
*and the latter is the area that contribute runoff during a mean annual runoff event (Mowchenko*
*and Meid, 1983). Thus, at its simplest, the non-effective area is the difference between the gross*
*and effective drainage area; however, the exact area contributing runoff is dynamic and the*
*controls complex, which include antecedent storage capacity and climatic conditions (Shaw et al.,*
*2012: Shook and Pomeroy, 2015)."*
Line 177. Please change the wording to "seasonally flooded prairie potholes". Potholes
are permanent landscape features, whereas flooded areas can be seasonal.
*Thank you for the clarification, and we have considered this comment in our revision. Given*
*suggestions made by Referee 2, we have adjusted the sentence to indicate what is meant be*
*"prairie potholes" as follows: "As such, "wetland" in this context can include some seasonal ponds*
*(i.e., prairie potholes) as well as larger or more permanent shallow water bodies".*
Line 180. Is (wetland density) needed here?
*We thank the reviewer for the suggestion. We removed this fragment and adjusted the sentence*
*for clarity.*
Line 191. Please briefly explain the meaning of mu and beta, and indicate the dimension
or unit. These must have a unit of area to maintain the dimensional homogeneity.
*We thank the reviewer for the suggested and the paragraph was modified to describe the*
*meaning of the Pareto distribution parameters and the units. The paragraph now provides*
*explanation of the meaning of the parameters within our context and the units.*
Line 195. Is it true that all pixels in the Canadian Prairie have "monthly" satellite images?
I do not think that is the case. Please clarify that in the texts.
*We thank the reviewer for their comments. The maximum water extents were computed from*
*Landsat images over the 32-year period, which have 8-day or 16 day revisit times. In this context,*
*the Canadian Prairies has monthly satellite images. We have removed the sentence of concern*
*and added the following for clarity: "Note that because the sizes of the water bodies were taken*
*from infrequent remote-sensing measurements (i.e., the Landsat data have a minimum revisit*
*time of 8 or 16 days), they also are biased against short-lived water bodies."*
Line 197. What do you mean by "the median area of the largest wetland"? Please
re-phrase so the reader can understand what you mean.

152   *We have clarified this in the text by adding more detail in the description of the term, as well as in*
153   *the Line of concern. It is the median of the distribution of the "area of the largest wetland" ($W_L$) for*
154   *the watersheds within each class. We provide the following description in the text: "The median*
155   *area of the distribution of largest wetlands for each watershed class provided an indication of the*
156   *maximum sizes of the water bodies exhibited those watersheds, and thus provided a maximum*
157   *value to simulate fitted values".*

Line 205. Surficial geology is mapped by geologists in each province using different
terminologies. I am not sure if the "comparison across provincial boundaries" is straight
forward. Please add a brief explanation on how the difference in terminology and
mapping methods was reconciled.

164   *Amelioration among surficial geology definitions was performed by grouping more defined*
165   *classification into broader categories describing depositional features. Grouping was performed*
166   *by comparing definition of each feature type using the provincial government metadata and*
167   *informed by advice from a colleague in geology. We acknowledge that these are broad groupings*
168   *and ideally we similar framework used across the provinces would be ideal. However, for our*
169   *current purposes, these broad descriptions were useful in capturing a variation in at least broad*
170   *geological settings.*

Line 208. In the Canadian System of Soil Classification, colour indicates more than just
an appearance of soil. For example, Black Chernozem and Dark Brown Chernozem
are distinct soil types developed under distinctively different climatic conditions. The
distribution of these soil types often coincides with ecoregions (e.g. Black Chernozem
is associated with Aspen Parkland). Please consult with local soil scientist to give a
better context to soil classes. Also, somewhere in the paper, perhaps near the beginning
of the method section, it will be useful to present a process-based framework to
understand the eco-hydrological functions of the Canadian Prairie landscape (see my
comment on Line 162).

182   *We thank the reviewer for this insight and have edited the text accordingly. We recognize that the*
183   *"colour" is only a descriptor and the function of the soils are different among soils types, and that*
184   *they develop under specific climatic conditions, geology, and vegetation. These were implicit in*
185   *the data that we used. We also included soil texture class data to provide additional description of*
186   *soil characteristics.*

Line 223. Please indicate the unit of DSF. It must be the inverse of length.

190   *We thank the reviewer for the comment. We adjusted the description to indicate that DSF is in*
191   *units of $km^{-1}$. We also added units for perimeter (km) and area ($km^2$).*

Line 255. Please indicate these prairie stations in Figure 5. I assume these are the
"study watersheds" described in Line 472. Please point that out here.

196   *We note the "study watersheds" in Line 473 is misleading. Here we are referring collectively to*
197   *the 4100+ watersheds used in the clustering analysis. We have revised the section for clarity.*

Line 265. Please explain how V1 and V2, and W1 and W2 are defined. Please note
that most readers of HESS are not familiar with CCA. You do not have to present
detailed explanation of CCA, but you need to give a brief outline so that the reader can understand the
basic concept.

204 *We thank the reviewer for the insight. We have made necessary adjustments to describe the*
205 *methods in more clarity. This concern was shared with the other reviewers. We have re-ordered*
206 *some of the sentences in the paragraph so that it now reads:*

208 *"Briefly, CCA involves correlating streamflow to physio-climatic characteristics of gauged*
209 *watersheds to create canonical variables. These canonical variables (i.e., V1, V2, W1 and W2)*
210 *are constructed from linear combinations of the original variables such that the correlation ($\lambda$) of*
211 *the canonical variables is maximized.  Positive canonical correlation coefficients imply positive*
212 *relationships and negative canonical correlation coefficients imply negative relationships.  There*
213 *are two canonical variable sets; one for physio-climatic variables (i.e., V1 and V2) and another for*
214 *hydrological variables (i.e., W1 and W2). Canonical variables plotting similarly on X-Y plots (W1-*
215 *W2 and V1-V2), indicate good correlation (Spence and Saso, 2005). If canonical correlation*
216 *values are above 0.75 (Cavadias et al., 2001), that set of variables was deemed useful for*
217 *estimating hydrological variables from physio-climatic ones. Those physio-climatic variables*
218 *passing this threshold were included as variables in a multiple regression to develop a predictive*
219 *equation for Q2. Analyses were performed using vegan package (Oksanen et al. 2018).*

221 Line 266. What are "the original variables"? Please explain, using a table if appropriate.

223 *We have adjusted the sentence for clarity by referring to the Table summarizing the original*
224 *variables.*

226 Line 290. ". . . attributes and is the basis . . ." for matching the tense.

228 *We thank the reviewer for the comment and have edited.*

230 Line 301. Please define alpha.

232 *We thank the reviewer for the comment and have edited.*

234 Line 310. What does this mean? Based on Line 269, does it mean that the result was
235 very useful for V1-W1, and barely useful for V2-W2? Please explain.

237 *We have adjusted the sentence for clarity by referring to the Table summarizing the original*
238 *variables.*

240 Line 311. What correlation value would indicate "strong"? Does it have a statistical
241 level of significance, like in the standard correlation analysis? Does a negative value
242 indicate negative correlation? Please explain.

244 *Thank you for the suggestions. Yes, positive correlation coefficients imply positive relationships*
245 *and negative correlation coefficients imply negative relationships. We have included these*
246 *descriptions to the methods description of the CCA, as included in the new paragraph above.*
247 *There is a sentence included that says "if correlation values are above 0.75 (Cavadias et al.,*
248 *2001), those were deemed useful for estimating hydrological variables from physio-climatic ones."*

250 Line 311-312. It is true that the correlation value is strong between Q100 (1:100 flow)
251 and W2, but it is weak for Q2 (mean annual flow) and W2. On the other hand Q2
252 and W1 has a strong correlation. Also the lambda value is much greater for V1-W1
253 combination than for V2-W2 combination. Given that, why was W2 chosen? Is it
254 because the classification is designed for 1:100 flood prediction? Please provide an
255 explanation.

*The second set of canonical variables (V2 and W2) were chosen because the individual*
*canonical correlation coefficients were higher than V1 and W1. We rephrase the paragraph to*
*discuss bias and reason for choosing the variables: "This sentence has been included into the*
*text: "The canonical coefficients from the CCA were λ1 0.97 and λ2 0.77, respectively. Mean*
*canonical correlation values between the hydrological variables and W2 were greater than those*
*with W1 (Table 1), and because both values of □ were acceptably large (Cavadias et al., 2001)*
*the physio-climatic variables strongly associated to V2 were used in the multiple regressions0 …*
*Plots of observed and predicted runoff Q2 (R2=0.45) and Q100 (R2=0.48) show moderate*
*agreement at lower flow values (Fig. 2). There is a negative bias estimated between 26 and*
*29%,….".*
Line 322. How is rock fraction area calculated? I cannot imagine there are many areas
of exposed bedrock in the Canadian Prairie. Please explain.
*There are regions of exposed bedrock, particularly in Southern Saskatchewan. We invite the*
*reviewer to the following map of surficial geology at*
[*http://publications.gov.sk.ca/documents/310/93756-*](http://publications.gov.sk.ca/documents/310/93756-Surficial%20Geology%20Map%20of%20Saskatchewan.pdf)
[*Surficial%20Geology%20Map%20of%20Saskatchewan.pdf*](http://publications.gov.sk.ca/documents/310/93756-Surficial%20Geology%20Map%20of%20Saskatchewan.pdf) *. Rock is shown in pink, and is*
*labeled "R". This landscape was mainly associated with dissected valleys and riverine systems.*
Line 326. Please list the classes of surficial geology used in the analysis.
*We have included a table of the surficial geology classes, as well as over components of the*
*compositional datasets, in the supplementary data (Table S3).*
Line 347. What are the "PCs from compositional datasets"? Are these different from
PC1-PC6 in the header of Table 3? Please explain.
*These are not the same Principal Components (PC). The "PCs from compositional datasets"*
*were used to capture the main gradients in the physiogeographical dataset (e.g., surficial*
*geology) that are then used in the PCA for the cluster analysis. This was comment was also*
*echoed by the second reviewer. We will evaluate how we explain our methods to increase clarity,*
*perhaps with added attention in the written methods section or inclusion of a figure that shows the*
*workflow.*
Line 358. "Weaker", not "less strong".
*We have revised accordingly.*
Line 389. The Canadian Prairie has now been divided into seven classes, which seem
to be consistent with our current understanding of eco-hydrology. For example, C1
roughly coincides with the ecoregion "Lake Manitoba Plain (162)" in the Ecozones and Ecoregions of
Canada (Ecological Stratification Working Group, 1995). Then, what
new knowledge and insights can we learn from this exercise? It will be nice to see a
clear demonstration of the contribution of this study to new advances in "Hydrology and
Earth System Sciences". Please try to present that in the discussion section.
*We thank the reviewer for their insights into the use of eco-hydrology and comparing our findings*
*to these classifications. We included references to ecoregions and discussed the similarities and*
*difference in these two approaches in the Discussion. Briefly, we see some relationships with*

*boundaries, however, we can identify areas that are not considered in the more general*
*ecoregion description, and provide a discussion on new insights gleaned beyond ecoregions.*
Line 412. Glacial till and hummocky landforms. Does this refer to one thing, or two
separate things (till and hummocky landforms)? Hummocky landform is a sub-class of
glacial till terrain. Please clarify.
*We thank the reviewer for this observation. It is true that hummocky landforms are associated*
*with glacial till deposits. However, the landforms dataset describes forms that include aspects of*
*surficial geology, relief, among others. Therefore the two datasets are related. We feel that both*
*datasets offer information on local geography. The hummocky landform designation is particularly*
*useful for characterizing landscape drivers depressional storage and overland flow.*
Line 453. Brown Chernozem is associated with the "Mixed Grass (159)" ecoregion,
which covers much of the driest part of the Canadian Prairies, commonly referred to
as the "Palliser Triangle". Accordingly the outer boundary of C5 roughly coincides
with the outer boundary of Mixed Grass. However, Figure 5 shows a patch of C6
in the core of the Mixed Grass, which is the driest part of Alberta having distinctly
different eco-hydrological characteristics compared to the band of C6 parallel to the
western boundary of the Prairie. Is the new method picking up new information, or is it
erroneously classifying watersheds? Are there too many classes in the system? These
are worth discussing in this section.
*Thank you for your observation. The classification indeed classifies watersheds outside of what*
*would be defined as a traditionally eco-hydrologically-based region. We expand on this idea in the*
*Discussion of our revised version. Briefly, we have confidence that the majority of watersheds are*
*being classified similarly resulting from our resampling analysis. Although some watersheds might*
*be seemingly spatially disparate, they exhibit characteristics that warrant membership to a*
*specific class. In the case of C5 and C6, they coincide well with the Mixed Grass ecoregion;*
*however they differ fundamentally in physical controls on hydrology (e.g., slope, non-effective*
*area), and thus provide additional information beyond ecoregion description.*
Line 472. Are there 11 study watersheds, as indicated in Line 255? If so, is that a high
enough number to examine all seven classes? Please explain.
*We address the concern with the miscommunication of the "study watersheds". However, we*
*acknowledge the concern of extrapolating data from 11 watersheds. However this is an*
*approximation of a hydrological runoff variable.*
Line 490-493. It is true that few studies have classified "watersheds" in the prairies,
but there have been numerous studies examining the spatial distribution of ecohydrological
functions of the Prairie landscape. For example, ecoregions are an integral
measure of hydro-climatology. Please acknowledge previous efforts and highlight the
newness of this work.
*We discuss this above. We added acknowledgement of the contribution of ecoregions in the*
*Discussion. We thank the reviewer for the insight.*
Line 502. This is an example demonstrating the strong effect of ecoregions on hydrology.

*We discuss this above and thank the reviewer for the insight. We added acknowledgement of the contribution of ecoregions in the discussion under section 5.1.2.*

Line 633. Yes, but the delineation has been available for many decades in the form of ecoregions. Please acknowledge it.

*Given the comments related to ecoregions, we have added a section within the discussion to discuss the similarities and differences it the approaches, and insights gleaned.*

Line 637. Geography may not be an appropriate term here, because geography encompasses many things, not just landforms. I would say topography or landform is more appropriate.

*We agree with this edits and the sentence has been revised to consider the comment. "Geography" was switched to "topography".*

Line 661. Figure 8 just shows wetland density and area delineated in satellite images, which is dependent of climatic factor (wetness) in addition to depressional storage capacity. Overall, I believe that the data from the 11 study watersheds can be utilized more to demonstrate the validity and usefulness of the new classification method. For example, are there distinct differences in the hydrological characteristics of seven classes of watersheds?

*As mentioned above, the 11 watersheds were only used for the CCA. The issue with using these to compare the classes is that these watersheds do not compare to the same scale as the watersheds derived from HydroSHEDs. Moreover, they tend to represent large, river-dominated systems, and mostly coincide with C4, C6, and C7. We use the wetland simulated data to compare how the classes represent observed data. We thank the reviewer for their comments, and we have elaborated on this in the text.*

**Response to Referee #2**

**Response to GENERAL COMMENTS**

*We appreciate the helpful suggestions and advice provided by Referee #2. Overall, the suggestions constructively added to the content of the manuscript. Specifically, we have added additional references and re-ordered the structure of the Introduction to emphasize applicability to an international audience. We also divided the Methods section into Data Collection (2) and Data Analysis (3) as per the suggestions of Referee #2. We felt this suggestion added to the readability of the manuscript. Finally, we have added more detail on the CCA method, which was a concern shared by other reviewers.*

**Response to SPECIFIC COMMENTS**

1. International readers might not be able to place the Canadian Prairie on a map (line 55). A brief statement about the geographical extent of the Prairies would help.

   *Increased detail regarding the Prairie region, and what distinguishes it, was also suggested by reviewer #1. As discussed in our response to reviewer #1, we provide greater detail of the Prairies ecozone in Canada in the methods and introduction, including the spatial extent of the region in the introduction.*

2. "Hydrological characteristics" (line 71) is unclear. Do the authors mean catchment attributes (e.g. topography, soils), climatic conditions, statistical properties of the streamflow regime or something else?

   *Yes, here we mean statistical properties of streamflow regime. This clarification has been added in the text.*

3. It would be helpful for the reader to briefly summarize how well earlier classification attempts have worked (line 74-78) and where the authors see current challenges.

   *In this regard, we are not concerned with whether these approaches have not "worked" but rather that although there have be attempts to classify watersheds/regions, they either do not extrapolate across provinces or are too coarse to represent heterogeneity within the Prairie. This is now better described in the Introduction. As reviewer #1 pointed out, ecoregions have been used to represent hydrological response by landscape characteristics in eco-hydrology. Our response to this latter comment can be found in our response to Referee #1. We appreciate the suggestion from reviewer #2 and provide detail to address some of this concern.*

4. The HydroSHEDS webpage (https://www.hydrosheds.org/page/development) lists a few regions where the data set is prone to errors, including areas with low or not well-defined relief. Is this of concern in the Canadian Prairies?

   *The error associated from datasets derived from SRTM can be of concern for the Prairies. Given this, the dataset does provide us with delineations at the scale of interest (~100km$^2$), and is the only dataset of that sort available. As a result, we deem it sufficient for our purposes given the current state of data availability for the region. We acknowledge the uncertainty in the dataset in the text with the following revision: "As with other SRTM products, the HydroSHEDs dataset may be prone to errors in regions with low relief due elevation precision of 1 m. However, the dataset*

*provided an objective delineation over the region of interest and was sufficient for purpose of the*
*current study."*

5. Approximately how many meters are 15 arc-seconds (line 140) in this area?

*This comment was shared with Referee #1 and we provide the distance measure in meters: "The*
*resolution is equivalent to for example approximately 285 m east-west and 464 m north-south at*
*Saskatoon, SK."*

6. What motivated the choice for these specific area (line 142) and urbanization (line
143, Table S1) thresholds?

*The choice in threshold areas was to remove very small "watersheds" or those that were very*
*large, which tended to relate to lake basins (e.g., Lake Winnipeg). The urbanization threshold was*
*informed by visual inspection of watersheds surround known large urban centers. A threshold of*
*40% removed most of those that had a large portion covered in urban development. We wanted*
*to focus on those watersheds that were more "rural" and reduce the immediate impact of cities or*
*development, which are known to produce unique impacts on local hydrology. We could not*
*remove urbanized areas completely due to the number of rural communities and roads that exist*
*across the Prairie region. We acknowledge the legitimate impact of cities and urbanization on*
*water quantity and quality necessitates consideration, but these questions are not in the scope of*
*the current manuscript. We added: "Because HydoSHEDs includes the basins of larger water*
*bodies, including lakes, watersheds consisting of majority water were removed as the study*
*concerns the uplands of these systems. Finally, highly urbanized areas (i.e., watersheds with*
*cover being >40% urban) were removed."*

7. The spatial resolution of climate data (line 157) seems large compared to the resolution
of the watershed boundaries. Can climate data on this resolution still be considered
representative for the smaller catchments?

*Please see related comment on the CANGRD in response to Referee #1.*
*The text now states that the original data has been interpolated by kriging to a higher spatial*
*resolution raster.*

8. What is the rationale for choosing the Thornthwaite method (line 161)?

*This comment was shared by Referee #1. The text now includes an acknowledgement of the*
*reason for choosing this method and a limitation: "To maintain consistency among climate data,*
*and use the same temperature data as described above, options were limited with which to*
*calculate PET. PET was calculated from the Thornthwaite equation (Thornthwaite 1948) using*
*the SPEI package (Vicente-Serrano et al., 2010). A disadvantage of the Thornthwaite approach is*
*it assumes a correlation between temperature and radiative forcing and adjusts for any lag in this*
*relationship using corrections for latitude and month."*

9. Snow formation and melt can strongly influence the seasonal water distribution
and accounting for the fraction precipitation that occurs as snowfall has recently
proved valuable in hydrologic similarity research (Knoben et al, WRR, 2018;
https://doi.org/10.1029/2018WR022913). Is there any particular reason why the authors
use only mean P and ET in their clustering?
*We thank the reviewer for the suggestion, and we agree that inclusion of this parameter is and*
*likely valuable for the Prairies. We focused solely on precipitation and ET because these*
*variables were available at the temporal length and spatial extent for the study. Given the*

*limitations of the dataset we used, calculating parameters at a seasonal scale might introduce*
*additional uncertainty, and thus was not included here. However, fraction of snowfall should be*
*considered in future iterations provide the data resolution is available.*
10. What is meant with a wet cycle (line 176-177)?
*We removed reference to a "wet cycle" and the sentence now reads: "The 30-year period was*
*chosen to capture natural climate variability". We thank the reviewer for their comment, and we*
*think this edit better reflects our intentions.*
11. Please include a (short) definition of potholes (line 177).
*Thank you for the comment. Given suggestions made by Referee 1, we have adjusted the*
*sentence to indicate what is meant be "prairie potholes" as follows: "As such, "wetland" in this*
*context can include some seasonal ponds (i.e., prairie potholes) as well as larger or more*
*permanent shallow water bodies".*
12. Why is the Lw/Lo metric (line 184) relevant? What does this metric tell us about
watershed behaviour?
*The metric identifies how close (or far away from) the largest wetland depression is to the*
*watershed's outlet. It is meant to be an indicator of hydrological gate-keeping and thus controlling*
*the likelihood for the watershed contributing flow to the downstream watershed. We explain this*
*concept in the Introduction and beginning of the Methods. We considered placing more context in*
*this regard, and we added the following clarification: "Both WL and LW/LO can be used to*
*evaluate the relative importance of hydrological gate-keeping; for example, larger wetland*
*depressions located closer to the outlet control the likelihood of the watershed contributing flow*
*downstream and attenuating peakflow (Shook and Pomeroy, 2011; Ameli and Creed, 2019)."*
13. The climate data (line 156), land cover data (line 230 and further) and hydrological
data (line 252 and further) cover different periods in time (1970-2000 for climate,
2011/2016 for agriculture land use, 1990-2014 for hydrologic data). For a general classification
of similar regions, overlapping time periods for the data sources would be more appropriate. What is the
rationale for not doing this?
*We think the reviewer offers a valid concern and we thank them for the insight. Land cover*
*because we wanted the most recent measurement to show current cover. The older climate data*
*was used because of the reduction in reliable precipitation data from Canadian climate stations*
*since 2000. Additional explanation of this now provided in the text.*
14. Estimation of mean flow Q2 and flood Q100 (line 252) for 4175 watersheds using
only 11 stations (line 255) seems ambitious to me. Spence and Saso (2005) show a
significant uncertainty in their predictions. Can the authors provide a statement about
their confidence in the Q2 and Q100 estimates?
*Spence and Saso (2005) evaluated uncertainty in predicting streamflow using canonical*
*correlation analysis and suggest that Q2 and Q100 estimates could exhibit errors of approaching*
*50% but exhibited bias of only 13%. We have elaborate on this topic in the text.*
15. What is the reasoning behind the 80% threshold for PCA components (line 279)?
Perhaps the authors can include a plot or table that shows the importance of each PC
to support this choice.

*The Scree plot in Figure 3 shows the importance of each PC in the analysis. The 80% threshold*
*is commonly used as a cut-off value for PCAs, which informed our decision how to limit PCs*
*considered for these dataset.*
16. Were variables standardized to a fixed interval (e.g. [0,1]) in addition to the logtransform
(line 282)?
*Fractional variables were standardized to a fixed interval because of the nature of the data.*
*However, other variables were not fixed (e.g., elevation).*
17. Line 286-287 needs clarification. Which variables are the "complete suite of variables"?
The previous section gives the impression that all variables were converted to
PCs, of which only those above 80% would be used. A table with a summary of all
variables used, their data source(s) and their hydrologic relevance could help clarify
what is going on.
*We recognize the vagueness of "complete suite". We have included the reference to Table 3 to*
*indicate the variable that were included in the analysis. The sentence now reads: "Clustering*
*analysis was performed on the complete suite of physio-geographic variables, which included PC*
*variables derived from pre-processing (Table 3)."*
18. Retaining PCs above 50% (line 291) seems to contradict retaining PCs above 80%
(line 279).
*The agglomerative clustering approach requires selecting the number of PCs included in the*
*analysis. This cut-off was chosen based on inspection of the contribution of PCs to the clustering*
*approach and described multiple co-related variables, rather than individual variables, which*
*tends to be the case for increasing PC number. This reasoning is why these two thresholds differ.*
*We have included the following with the intention of being clearer: "Retaining these first PCs at a*
*threshold of 50% allowed for clearer focus on main trends in the data and reduced the impact of*
*noise on subsequent analyses, which might occur if subsequent, less influential, PCs were*
*retained."*
19. A short description of Ward's criterion (line 295) would be helpful.
*Thank you for the suggestion. We added additional description as follows: "Ward's criterion*
*decomposes the total inertia of clusters into between and within-group variance, and this method*
*dictates merging for clusters (or watersheds) such that the growth in within-group inertia is*
*minimal (Husson et al. 2010). Within-group inertia represented the homogeneity, or similarity, of*
*watershed within a cluster."*
20. I suggest replacing "and thus did not explicitly affect the clustering analysis" (line
303) with "and are not included in the clustering procedure" (assuming that I correctly
interpreted this sentence).
*Variables included in the analysis as "supplementary" had their relative location in PCA-space*
*calculated (i.e., eigenvalues were calculated for the variable for each PC). However, they did not*
*impact the PCA directly, which is in contrast to "active" variables. The suggested revision is not*
*completely accurate; we have adjusted our original explanation to mitigate confusion. We have*
*include the following sentence, which is now in the previous paragraph to denote that this step*
*occurred before the HCPC: "The majority of physiogeographic variables were included as active*

*variables in the PCA and thus influenced the arrangements of the PCs. In contrast, watershed area, DSF, latitude, and longitude were used only as supplementary variables, and thus did not explicitly affect the clustering analysis. These variables did, however, aid in watershed class characterization and interpretation."*

21. Not all readers will be equally familiar with canonical regression analysis. I find it difficult to interpret the results in section 3.1. A (very) brief description of CCA might help. Some questions I'm stuck with: are those lambda values high or low? What would either tell us? What does it mean that hydrologic variables are associated with W2?

*We provided more detail in regards to the CCA method and include references where necessary. This concern was shared by other reviewers.*

22. I would say these regressions are not particularly convincing (line 314). It looks as if the one high value could be inflating the correlation value. Did the authors use Pearson or Spearman correlations? Predicting streamflow characteristics in ungauged basins (i.e. regionalization) is an active field of study but achieving robust results has proven very difficult. How does this impact the extrapolation of this information to the 4100+ watersheds and what are the consequences for the subsequent analysis?

*The bias in this relationship is 29 – 26 %. Perhaps this is to be expected give the small sample size. It is higher than that documented by Spence and Saso (2005) in their study. Content to this point has been added to the manuscript.*

23. Section 3.2 (PCA results) lacks a logical conclusion (or perhaps an introduction). How did the authors choose how many PCAs to discuss and which PCAs are selected to be used in subsequent steps?

*We intend for this section to provide an account of the main variables associated with the PCs of the compositional dataset. We see these as intermediate steps within our procedure and is intended to provide a brief overview of this preliminary step. We thank the Referee for the suggestion. We have provided elaboration on the clustering PCA as per comment #25 to increase clarity.*

24. The difference between active and supplementary variables needs to be defined (line 348).

*Thank you for the suggestion. We have clarified the difference between active and supplementary variables in the Methods section as per comment #20.*

25. Section 3.3 lacks a logical conclusion. Which PCAs are carried over to the clustering analysis?

*The intention of this section was to describe the PCs and the variables associated with them. We considered it an intermediate step within our procedure, and the 6 PCs were used in the following clustering analysis. We appreciate the reviewers comment, and added sufficient detail to strengthen the relationship between this step and the cluster analysis. This includes a paragraph outlining trends and important characteristics briefly, followed by a more detailed account on the relationships of individual parameters to each principal component. We have also provided a figure in the supplementary material displaying our workflow to improve clarity (Fig. S1).*

26. What do the authors mean with "definition of clusters" (line 370)?

*Here, "definition" refers to the distinction of each class. We adjusted the sentence to read:*
*"Further increasing k improved definition refined the separation and definition of clusters up to*
*seven (k=7)."*

27. Section 3.4 is very brief. One of the main aspects of clustering analysis is assessment
of how good the resulting clusters are. Currently the authors extensively list the
differences between the clusters (section 3.5) by summarising which inputs were most
influential in determining the clusters. However, this only tells us something about
the patterns in the data and not much about the usefulness of these clusters. The
authors suggest in the discussion that these clusters can be helpful to inform management
decisions, by showing which regions are expected to behave similarly and
which regions are not. This statement should be backed up by proof with independent
data that these cluster indeed show that. The GSIM archive (Do et al, HESSD, 2018;
https://doi.org/10.5194/essd-10-765-2018) is a recent contribution of global streamflow
indices which might provide the authors with independent hydrologic information that
they can use to quantify how well their clusters group hydrologically similar regions.
See e.g. Knoben et al, WRR, 2018 (linked above) for possible ideas.

*We thank the reviewer for this insight. Comparison with independent data was also suggested by*
*Referee #1. We elaborate on this comment at the beginning of our response. We have also*
*included another analysis that compares the robustness of the clustering approach. In addition,*
*we evaluate the applicability of some independent data sources, (e.g., HYDAT, wetland remote*
*sensed data) to compare our classes and the appropriateness of their use, in our responses*
*above and in our Introduction. We also further incorporate the comparison with simulated and*
*observed wetland size distributions. Our intention here is to compare how the classes represent*
*the observed data of the watersheds within each sub region. Streamflow data (from Do et al.*
*2018) is likely not appropriate for most of the watersheds classes and are not available at the*
*spatial and temporal resolution necessary; although we appreciate the reference to this work. We*
*use the wetland dataset for this purpose. Despite the limitation within these remotely sensed*
*data, we feel it provides a useful application to the prairie regions as well as those regions that*
*are semi-arid or do not possess a well-developed drainage area where streamflow comparisons*
*are not representative.*

28. The subsections of section 3.5 are hard work for an international audience.
Perhaps figure 5 can be expanded to include a map which shows the various
names used in these sections (see e.g. Addor et al, HESS, 2017; figure 1e;
https://doi.org/10.5194/hess-21-5293-2017 )

*We thank the reviewer for their insights regarding readability for an international audience. We*
*point to Fig. 1 for reference to the Provincial names. We also removed reference to more specific*
*and local landmarks (such as Quill and Manitou Lakes). We keep references to the major rivers*
*within this region.*

29. Line 435-437 ("Being river valleys . . . Q2 values (Table 1)) repeats line 428-429.

*Thank you for the comment, we have removed the repeated line.*

30. I'm unsure how section 3.6 relates to the previous clustering results. I was under the impression that wetland density is one of the variables used during clustering.
Should section 3.6 perhaps be moved to before the clustering results? Also, if this
is part of the clustering analysis (as e.g. table 3 and 4 seem to suggest), why does
this specific attribute deserve its own section? Edit: reading back, it seems to me
that wetland distributions were estimated (line 186 and further). In that case, are the
observations referred to in line 480 from the 11 stations? This seems a small sample
of observations to compare results for 4100+ watersheds to. How confident can we be
in these estimates?

*The simulated wetlands by class shown in section 3.6 (Figure 8c) were calculated based on the*
*Generalized Pareto Distribution (GPD) parameters ($\xi$ and $\beta$) that were used in the clustering*
*analysis. The wetland density and $W_L$ parameters in panels (a) and (b) were discussed to provide*
*context to the simulated data in panel (c). To clarify, the observed quantiles were based on those*
*from each of the 4100+ wetlands, and the predicted values were from the simulated data based*
*on the GPD parameters. Our intention was to provide an example of how the classes translate to*
*observed data, which is consistent with reviewer suggestions that such an approach could*
*strengthen the study. Specifically, we can predict wetland size distributions from the parameters*
*in the classification, and that the simulated data is relativity consistent with the observed data. We*
*elaborate on the usefulness of these data and our intentions in the discussion. We have also*
*added section 3.4 and 4.4 to be clearer in our intention for this comparison.*

31. The authors stress the importance of accounting for human influences (Section 4.1)
in classification procedures. Can they comment on the extent to which this was done
in their work and do they have any recommendations for future efforts? For example,
should artificial drainage density be considered as a variable?

*In this regard, data availability at the appropriate geographic scale and spatial resolution is*
*limiting, as we indicate in the text. We incorporate human dimension to a degree, with the*
*inclusion of tillage practices and area of land cropped. Artificial drainage density would be a very*
*useful indicator; however, a comprehensive dataset is not available for the region of interest. We*
*plan to pursue avenues for including a proxy for this parameter in the future. We discuss the*
*usefulness of an artificial drainage estimate in line 675.*

32. The authors mention that certain variables can dominate the clustering approach
(line 579 and further). This is why it is not uncommon to standardize clustering variables
to a fixed interval, because this reduces the effect of a variable's variability.
Log-transforms lessen, but do not prevent this. Can the authors comment on which
variables had the widest (log-transformed) range and whether this correlates with the
variables that are most important during clustering?

*Thank you for providing the suggestion to compare the impact of fixing variables to an interval.*
*Scaling variables during the PCA was performed in our procedure, which might help to address*
*this concern. In this particular case, such as the fraction of watershed below the outlet, we*
*indicate that despite hydrological importance, a couple variables might not have been indicated*
*as important to characterizing the classes. Our discussion attempted to elude potential*
*overshadowing that might occur. Moreover, if one is particularly interested in such variables, one*
*should consider strategies to weight their importance. It should be noted that the fraction below*
*the outlet was an important variable for Class 5, just that it was not consider highly important to*
*the other classes amongst the various other competing characters. We have adjusted our*
*Discussion section to be clearer in this regard.*

**Response to Referee #3**

*Please see below for point-by-point comments to Referee #3's suggestions:*

Ambiguity: It has been mentioned that the CCA was used for estimating hydrologic variables since only a few observing stations are available. These variables will be considered later in the classification approach to provide a watershed classification system that will be used, among other purposes, to estimate the hydrological response of a given watershed. What is confusing and contradicting here is to first estimating hydrological variables, and then using classification outputs to understand the hydrological behavior! A regionalization approach is more suited for this purpose.

> *In order to reduce the ambiguity we have rewritten this section.  The second paragraph now reads:*
>
> *To address this gap mean annual runoff and 1:100 year flood magnitude had to be estimated for each of the 4175 watersheds.  Canonical correlation analysis (CCA) was used for this purpose because it was felt that it provided a more independent means of regionalization than using terms directly applied within the subsequent cluster analysis.   CCA was used to correlate gauged data to ……"*

I feel inconsistency in using CCA (the most appropriate classification method as recognized in regionalization studies) to estimate hydrological variables, and using another classification method, hierarchical cluster analysis, for classification.

> *As stated above, we needed a method to obtain streamflow terms for each of the 4175 watersheds that was somehow more independent.  We believe we have explained why we needed to use a regionalization method to estimate Q2 or Q100, but the objective of the study was to classify the watersheds, and the hierarchical cluster analysis is a more appropriate tool.*

Equation in Line 319 is not very convincing since no precipitation or water-related variable is introduced.

> *One is not necessarily required.  The canonical correlation coefficients imply Q2 can be estimated with confidence using these terms and with the values in the equation.*

Also, only 11 observations have been considered for calibration. Assessment of the uncertainty is not consistent too.

> *We felt an uncertainty assessment of the equation in Line 319 was unnecessary because of how the estimate of Q2 was used.  To do so would have meant an uncertainty analysis could have been required for every other input into the cluster analysis, which was beyond the scope of the paper.*

[revised manuscript text omitted]

---

## Referee Report (RR1)

**Review 28-05-2019**

A WATERSHED CLASSIFICATION APPROACH THAT LOOKS BEYOND HYDROLOGY: APPLICATION TO A SEMI-ARID, AGRICULTURAL REGION IN CANADA

Jared D. Wolfe, Kevin R. Shook, Chris Spence, Colin J. Whitfield

I have read the authors' response to reviewers and their revised manuscript with interest. To me the revisions seem thorough and the manuscript has become much clearer as a result. I commend the addition of a sensitivity analysis of the clustering approaches in section 4.4. The editor has requested special attention to original comment 27, in which I ask the authors to evaluate the usefulness of their classification with independent data. This is addressed in point 6 in this review. My comments use the line numbers in the author response document (hess-2018-625-author_response-version1).

**Response to reviewers**

1. L430 (comment 5): the authors provide more detail about the accuracy of HydroSHEDs and state that "[…] the dataset provided an objective delineation over the region of interest and was sufficient for purpose of the current study." This argument would gain in strength if the authors can add how they came to this conclusion.

2. L474 (comment 8): the authors provide additional context for choosing the Thornthwaite PET method (which I think is justified) and also state a disadvantage of the method. It might be helpful to also include the practical impact of this disadvantage, because I don't quite understand.

3. L484 (comment 9): the authors provide reasons for not using any metrics related to snow in their study but acknowledge that this might be important. Is this mentioned anywhere in the manuscript? For example as a study limitation or an opportunity for further work.

4. L532 (comment 14): the authors provide a statement about the accuracy of the findings of Spence and Saso (2005). Is this accuracy dependent on the number of observations used? (Spence and Saso (2005) seem to use n = 34, compared to n = 11 in this paper). Addition: I see the authors have clarified this on L1267.

5. L555 (comment 17): the changed text in this response refers to Table 3, but the text in the manuscript refers to Table S3 (L1179).

6. L656 (comment 27): the authors provide evaluation against another data source (section 4.3) and include a sensitivity analysis of their clustering approach (section 4.4). The also point out (as reviewer #1 has mentioned) the rough correspondence between their clusters and the current understanding of eco-regions (section 5.1.2). The authors also comment that further evaluation is difficult due to the lack of data sources (e.g. L678).
   The authors state that (L928) "… those areas that are climatically and physio-geographically similar, and thus might be expected to respond in a hydrologically coherent manner to climate and land management changes." This is the critical assumption that underlies this clustering exercise. As I understand the manuscript, section 4.3 does not as much evaluate the entire classification, but only a part of it (wetland density). Further demonstrating that the defined clusters indeed respond in a coherent manner will add much more weight to this paper. However, if there is no data available than that is clearly not an option.
   If this is the case, the authors might want to further highlight the novelty of their work compared to the current understanding of eco-hydrology on the Prairies (e.g. the need for fuzzy treatment of watershed similarity as evidenced by section 4.4; the increased granularity possible with an approach such as the authors use, …).

7. L733: the authors provide more detail about how they scaled variables during the PCA. I'm bringing this point up again in relation to the text on L1326: "Climate and elevation gradients are likely responsible for the west to east watershed clustering pattern." I wonder to what extent this is forced by the data preparation, where these variables are log-transformed but not normalized. Is it possible that the log-transformed range of climate and elevation variables spans a wider range than that of the other variables? For example, if log-transformed mean precipitation has range [0,3] (assuming P = 1 to 800mm) it would span three times the range of a fractional variable with range [0,1]. This might skew the clustering procedure towards treating P and elevation as more distinctive attributes for each cluster. I don't believe this is necessarily a bad thing, for example if there are reasons to believe that P and elevation are relatively important. However, the authors also comment that "[…] if one is particularly interested in such variables, one should consider strategies to weight their importance." Is it possible that some form of weighting has already happened in the current manuscript as a result of only log-transforming the variables?
Investigation of the log-transformed ranges of each variable might indicate this. This would be a relatively low-effort check compared to re-doing the full clustering analysis with differently prepared data. If found relevant, this might be added to the discussion in L1595-1602.

**Comments on revised manuscript**

8. L877: "regime" > "regimes"?
9. L1033: I think the term "wet climate cycles" might still be confusing. Would "wet climate periods" be a suitable alternative?
10. L1208: it might be helpful to the reader to briefly summarize why wetland area distributions are simulated, if observations are also available (in the GSW data set). Am I correct in saying the GSW only gives the maximum wetland area, and the GPD simulation gives estimates of the full distribution of wetland sizes?
11. L1224: "4175 set" > "set of 4175 watersheds"
12. L1299: "TPC3" > "PC3"
13. L1425: "[…] less than 1." > "[…] less than 1 $km^{-2}$"
14. L1652: This paragraph might be better placed directly after (or as part of) the paragraph that starts on L1610.
15. L1683: Is the reference to Wagener et al, 2007 correct? I don't believe that paper talks about the relation between management practices and classification approaches. Perhaps this should be Wagner et al, 2007 (which I haven't read but its title suggests it as being more likely)?
16. L2127, Figure 6b: the number of points make this plot difficult to read. x-axis could be changed to cover the width of the page. Possibly cut of the y-axis at 10 for additional clarity.
17. Figure S1, c: text in the centre overlaps and is unreadable.

---

## Author Response (AR2)

**Response to reviewers: "Watershed classification for the Canadian Prairies"**
*Please note that we have changed the manuscript title to: "A WATERSHED CLASSIFICATION*
*APPROACH THAT LOOKS BEYOND HYDROLOGY: APPLICATION TO A SEMI-ARID,*
*AGRICULTURAL REGION IN CANADA".*
Page and line numbers are shown for each change in reference to the marked-up version.
**Response to Referee #1**
**Response to SPECIFIC COMMENTS**
1. L430 (comment 5): the authors provide more detail about the accuracy of HydroSHEDs and state that
"[…] the dataset provided an objective delineation over the region of interest and was sufficient for
purpose of the current study." This argument would gain in strength if the authors can add how they came
to this conclusion.
*We thank the reviewer for drawing our attention to this comment. Our conclusion was based on*
*data availability that both covered the geographic scale and resolution (i.e., 100 km$^2$) necessary*
*for the purposes of our study. However, in light of the reviewer's comments, we added clarity to*
*this sentence:*
*"However, the dataset watershed delineations over the geographic region of interest and at a fine*
*enough scale (i.e., 100 km$^2$), and thus, it was sufficient based on data availability for purpose of*
*the current study." (7, 174)*
2. L474 (comment 8): the authors provide additional context for choosing the Thornthwaite PET method
(which I think is justified) and also state a disadvantage of the method. It might be helpful to also include
the practical impact of this disadvantage, because I don't quite understand.
*We thank the reviewer for raising this concern, which was also raised by Referee #2. We have*
*added the following to increase clarity on the impact of this method and assumption:*
*"A disadvantage of the Thornthwaite approach is that it calculates PET solely as a function of air*
*temperature and latitudinal position, and it assumes a fixed correlation between temperature and*
*radiative forcing. As such, it integrates effects of other factors directly or indirectly influencing*
*radiation or latent heat, like advection, vegetation, and humidity. The calculation adjusts for any*
*lag in this relationship using corrections for latitude and month; however, it likely does not*
*represent the full annual and seasonal variability in PET across a landscape, given regional*
*heterogeneity of the aforementioned factors. Despite the limitations, the simplicity of this method*
*is ideal for application across the wide geographic area of interest with limited data required as*
*input, allowing for approximation of mean annual PET for the study area." (8, 209)*
3. L484 (comment 9): the authors provide reasons for not using any metrics related to snow in their study
but acknowledge that this might be important. Is this mentioned anywhere in the manuscript? For
example as a study limitation or an opportunity for further work.
*We greatly appreciate the reviewer's comments in regard to the consideration of snow variables.*
*We now reference in the Discussion the limitation of the current study in this regard, and that if*

*data is and becomes available, it should be included, or considered, in future classification*
*approaches.*
*"Where data is available, future work should consider variables related to snow formation and*
*melt, as well the proportion of annual snow to rainfall as these variables are likely influential when*
*describing hydrological behaviour of the watersheds and classes (Knoben et al., 2018; Shook and*
*Pomeroy, 2012). (28, 810)*
4. L532 (comment 14): the authors provide a statement about the accuracy of the findings of Spence and
Saso (2005). Is this accuracy dependent on the number of observations used? (Spence and Saso (2005)
seem to use n = 34, compared to n = 11 in this paper). Addition: I see the authors have clarified this on
L1267.
*We appreciate the reviewer's comment here. We do expect an impact on the uncertainty based*
*on the smaller sample size used in the current study. As the reviewer indicates, we clarify this*
*expectation when reporting results. To help with this concern, we clarify this expectation in our*
*methods: "We note that greater uncertainty than that reported by Spence and Saso (2005) may*
*result when using the CCA approach with a smaller sample size." (12, 335)*
5. L555 (comment 17): the changed text in this response refers to Table 3, but the text in the manuscript
refers to Table S3 (L1179).
*The reference should be to Table S3, which shows the compositional datasets. We also refer to*
*Table S2 which includes the source datasets. We have made the revision to say "[…] from pre-*
*processing (Table S2; Table S3)". (13, 371)*
6. L656 (comment 27): the authors provide evaluation against another data source (section 4.3) and
include a sensitivity analysis of their clustering approach (section 4.4). The also point out (as reviewer #1
has mentioned) the rough correspondence between their clusters and the current understanding of eco-
regions (section 5.1.2). The authors also comment that further evaluation is difficult due to the lack of data
sources (e.g. L678). The authors state that (L928) "… those areas that are climatically and physio-
geographically similar, and thus might be expected to respond in a hydrologically coherent manner to
climate and land management changes." This is the critical assumption that underlies this clustering
exercise. As I understand the manuscript, section 4.3 does not as much evaluate the entire classification,
but only a part of it (wetland density). Further demonstrating that the defined clusters indeed respond in a
coherent manner will add much more weight to this paper. However, if there is no data available than that
is clearly not an option. If this is the case, the authors might want to further highlight the novelty of their
work compared to the current understanding of eco-hydrology on the Prairies (e.g. the need for fuzzy
treatment of watershed similarity as evidenced by section 4.4; the increased granularity possible with an
approach such as the authors use, …).
*We appreciate the attention to comment given by the reviewer and editor. The lack of*
*hydrological data available (and thus that available for adequate validation) at the granularity and*
*spatial consistency was one of the motivating intentions of this study. Although only*
*representative of the part of the hydrological response, we show the differences in wetlands size*
*distributions of the classes in Figure 8. Given the relationship with wetlands and hydrological*
*response (citations therein), we also recognize the comparison suggests only potential coherent*
*difference in response. Future work intends to build on the foundation laid in this study and*
*compare the coherent hydrological responses to environmental change; however, we believe*

*including this analysis in addition to the current study would make the manuscript quite unwieldy.*
*We agree with the reviewer's suggestion to highlight the novelty in our approach, specifically the*
*scale and "fuzzy" boundaries, and we emphasize this in our discussion, such as the following*
*addition: "Our results are novel in that they characterize in greater detail, and at small watershed*
*scales, the potential for different hydrological behaviour of watersheds within the region." (23,*
*655)*
7. L733: the authors provide more detail about how they scaled variables during the PCA. I'm bringing
this point up again in relation to the text on L1326: "Climate and elevation gradients are likely responsible
for the west to east watershed clustering pattern." I wonder to what extent this is forced by the data
preparation, where these variables are log-transformed but not normalized. Is it possible that the log-
transformed range of climate and elevation variables spans a wider range than that of the other
variables? For example, if logtransformed mean precipitation has range [0,3] (assuming P = 1 to 800mm)
it would span three times the range of a fractional variable with range [0,1]. This might skew the clustering
procedure towards treating P and elevation as more distinctive attributes for each cluster. I don't believe
this is necessarily a bad thing, for example if there are reasons to believe that P and elevation are
relatively important. However, the authors also comment that "[…] if one is particularly interested in such
variables, one should consider strategies to weight their importance." Is it possible that some form of
weighting has already happened in the current manuscript as a result of only log-transforming the
variables?
Investigation of the log-transformed ranges of each variable might indicate this. This would
be a relatively low-effort check compared to re-doing the full clustering analysis with
differently prepared data. If found relevant, this might be added to the discussion in L1595-
1602.

*We thank the reviewer for this suggestion. We performed the log-transformed range check*
*suggested by the reviewer for each input variable. Upon observation, there does not seems to be*
*a relationship between the log-transformed range and those variables that were influential on the*
*classification procedure. Interestingly, Elevation and Precipitation had relatively low log-transform*
*range (1.8 and 0.8, respectively) compared to other log-transformed variables. It should be noted*
*that because we used annual precipitation, the range in our data would not be between 1 to*
*800mm. We do reiterate our previous response to this concern in that variables were scaled*
*when the PCA was performed. Our point in the discussion (26, 765) is to indicate that perhaps*
*approaches should consider that some variables that are particularly impactful on a local scale*
*(like the location of the largest pond), and that considering weighting might be a strategy to have*
*a hierarchy in what variables might be considered more important. However, we recognize that a*
*drawback to this approach is to increase the amount of assumptions one makes about the data*
*prior to data analysis. We have added the following to indicate that variable ranges were scaled*
*during the PCA: (1) "Variable unit ranges were also scaled during the PCA to reduce the impact*
*of certain variables exhibiting a large range of values on the subsequent cluster analysis." (13,*
*365); and (2) "Variable importance in the classification was not related the log-transformed range*
*exhibited by that variable (data not shown), and impact was mitigated by scaling the ranges of*
*input variables in the PCA." (17, 478)*

**Comments on revised manuscript**
8. L877: "regime" > "regimes"?
*We have made the edit (4, 83).*

9. L1033: I think the term "wet climate cycles" might still be confusing. Would "wet climate
periods" be a suitable alternative?
*We agree that "wet climate periods" is suitable. We have made the revision (9, 242)*
10. L1208: it might be helpful to the reader to briefly summarize why wetland area distributions
are simulated, if observations are also available (in the GSW data set). Am I correct in saying
the GSW only gives the maximum wetland area, and the GPD simulation gives estimates of
the full distribution of wetland sizes?
*The wetland distributions were simulated to compare how the use of the Generalized Pareto*
*parameters reflect the observed data based on the GSW dataset. Our simulations were restricted*
*at the lower part of the distribution to reflect the data resolution of the satellite-based data from*
*GSW (we reference this for example in P29, L853). Therefore, these simulations do not predict*
*the "smaller wetlands". However, our results suggest that parameters might be used in future*
*studies to predict across the distribution of wetlands, and they are useful parameters to describe*
*watershed or class wetland size distributions (P29, L857). We have adjusted the section of*
*concern as well to reflect this (14, 398).*
11. L1224: "4175 set" > "set of 4175 watersheds"
*Thank you. We have made the edit (15, 416).*
12. L1299: "TPC3" > "PC3"
*We thank the reviewer for the edit, and we have made the change (17, 494).*
13. L1425: "[…] less than 1." > "[…] less than 1 km-2"
*We have made the edit (21, 611).*
14. L1652: This paragraph might be better placed directly after (or as part of) the paragraph that
starts on L1610.
*We appreciate the suggestion and agree that the discussion on boundaries and analysis should*
*flow accordingly. We have moved the paragraph ahead (27, 796).*
15. L1683: Is the reference to Wagener et al, 2007 correct? I don't believe that paper talks about
the relation between management practices and classification approaches. Perhaps this
should be Wagner et al, 2007 (which I haven't read but its title suggests it as being more
likely)?
*We appreciate the comment. The reference should indeed be for Wagner et al. 2007. We have*
*made the change (30, 894).*

16. L2127, Figure 6b: the number of points make this plot difficult to read. x-axis could be
changed to cover the width of the page. Possibly cut of the y-axis at 10 for additional clarity.
*Thank you for the suggestions. We have adjusted the width of the figure and transparency of the*
*points in Figure 6b. Transparency was adjusted to show overlap among the points. We also cut*
*the figure at 10 as the trend remains relatively unaltered (two watersheds with densities higher*
*than 10 km$^{-2}$ are not visualized).*
17. Figure S1, c: text in the centre overlaps and is unreadable.
*The soil zones (Dark gray and gray) in this part of the plot are not represented well by Axis 1 and*
*Axis 2, which is the reason for the overlap. To help with interpretation, we have added to the*
*figure description: "Note that the variables at the centre of plot (c) are "dark gray" and "gray" soil*
*zones and are not well represented by Axis 1 and Axis 2."*

**Response to Referee #2**

**Response to GENERAL COMMENTS**

The authors have addressed many of my comments on the original manuscript, but did not present adequate response to a number of comments. I have annotated a PDF file (attached) of their response with further comments and suggestions. In addition, in Line 862 and 1510 in the marked manuscript, Hayashi and Rosenberry (2002) is cited as a reference regarding ecoregions, but this paper did not discuss ecoregions at all. Please remove the reference from these sentences.

*We thank the reviewer for noting this concern. We have removed the references for the respective lines.*

**Response to SPECIFIC COMMENTS**

Line 117. How is the Canadian Prairie defined? Please present a brief definition, and the source of the ecozone boundary shown in Figure 1.

Comments: I do not see the source in Figure 1.

*We have added the citation to the figure for the ecozone boundary.*

Line 136-138. As it is written, the sentence indicates that the watershed of the Saskatchewan River is excluded from the analysis, which is clearly not the case.

Comment: This has not been done. Please address the comment.

*Upon revision, we deemed that this clarification was not needed for describing which watersheds were included in our study region (i.e., those that fall within the ecozone boundary). We have removed the portion of concern, and the text now reads: "Thus, we constrained our study to the Canadian Prairie ecozone (4.7 x 10⁵ km²) and watersheds occurring therein." (7, 166)*

Line 141. The authors describe watersheds by referring the reader to Figure 1. However, Figure 1 does not show watersheds. Please refer the reader to Figure 5 instead, or add watershed boundaries to Figure 1.

Comment: The reference to Figure 1 has not been removed. Please show the watershed boundaries in Figure 1.

*Here, the reference to the figure means to address the extent of the Canadian Prairie ecozone, not the watershed boundaries. We have removed the reference here for clarity, as it was determined unneeded (7, 177).*

Line 161. Temperature-index methods such as Thornthwaite do not give reliable estimates of "potential evapotranspiration" … please explicitly acknowledge its limitation.

Comment: This sentence is unclear. Please acknowledge more specifically the well-known bias and error in PET estimates using the Thornthwaite and similar temperature-based methods.

*We thank the reviewer for raising this concern, which was also raised by Referee #1. We have*
*added the following to increase clarity on the impact of this method and assumption:*
*"A disadvantage of the Thornthwaite approach is that it calculates PET solely as a function of air*
*temperature and latitudinal position, and it assumes a fixed correlation between temperature and*
*radiative forcing. As such, it integrates effects of other factors directly or indirectly influencing*
*radiation or latent heat, like advection, vegetation, and humidity. The calculation adjusts for any*
*lag in this relationship using corrections for latitude and month; however, it likely does not*
*represent the full annual and seasonal variability in PET across a landscape, given regional*
*heterogeneity of the aforementioned factors. Despite the limitations, the simplicity of this method*
*is ideal for application across the wide geographic area of interest with limited data required as*
*input, allowing for approximation of mean annual PET for the study area." (8, 209)*
Line 162. The balance between precipitation and evapotranspiration is reflected in ecoregions of the
Prairie, as plants are good indicator of long-term water balance. … Please provide an explanation.
Comment: This response is missing the point. Ecoregions are defined by the optimal vegetation
community reflecting the climatic condition, not the actual land use and agriculture. Please present a
more meaningful response.
*We recognize that ecoregions are defined by the vegetation community, which results from*
*climatic conditions. We instead used data from the CANGRID product to approximate the water*
*balance across the region. Using the ecoregions to estimate the long-term water balance, while*
*an appropriate method to use, would confine these estimates to the respective boundaries of the*
*ecoregions. We aimed to provide an analysis independent of the pre-defined ecoregions (but see*
*our discussion in section 5.1.2). Using the CANGRID product allowed for gradients of both*
*precipitation and PET to be approximated, which we deemed preferable for the purpose of*
*classifying sub-regions of watersheds.*
Line 191. Please briefly explain the meaning of mu and beta, and indicate the dimension or unit. These
must have a unit of area to maintain the dimensional homogeneity.
Comment: Please indicate the dimension of beta.
*We explain the meaning behind the scale ($\beta$) and shape ($\xi$) parameters as well as their units (10,*
*259-264). We have replaced "scale" and "shape" with the Greek letter notation, respectively, for*
*clarity.*
Line 205. Surficial geology is mapped by geologists in each province using different terminologies. I am
not sure if the "comparison across provincial boundaries" is straight forward. Please add a brief
explanation on how the difference in terminology and mapping methods was reconciled.
Comment: Please add the explanation for this procedure in the texts.
*We appreciate the request of more detail in this regard. We added clarity to the text, and it now*
*reads: "Due to the different geological classification schemes for each province, more detailed*
*classes were grouped to broader categories related to depositional environment and surficial*
*materials using those from the Geological Survey of Canada (2014), which provided for*

*comparison across provincial boundaries." (11, 291). We also added this citation to the*
*references section.*
Line 266. What are "the original variables"? Please explain, using a table if appropriate.
Comment: I do not see the reference to the table. Please address the comment.
*Here, we are briefly describing the CCA method, and the "original variables" refer to those*
*physico-climatic and hydrological input in the analysis. As such, we rescind the reference to the*
*Table 2. We have adjusted the sentence to: "Briefly, CCA correlates the streamflow record of*
*gauged basins to physico-climatic characteristics of watersheds by representing these variables*
*as a reduced set of canonical variables. The analysis results in two canonical variable sets: one*
*for the physico-climatic variables (i.e., V1 and V2) and another for the hydrological variables (i.e.,*
*W1 and W2). These canonical variables are constructed from linear combinations of the variable*
*sets such that the correlation of the canonical variables are maximized." (12, 343)*
Line 301. Please define alpha.
Comment: I do not see the definition. Please address the comment.
*The term α refers to the level in which statistical significance was determined. We removed this*
*reference to alpha and replaced it with "$p < 0.05$". (14, 395)*
Line 310. What does this mean? Based on Line 269, does it mean that the result was very useful for V1-
W1, and barely useful for V2-W2? Please explain.
Comment: The adjusted sentence does not address this comment. Please make a more meaningful
adjustment.
*We thank the reviewer for their feedback. We have re-arranged the sentences with the aim of*
*adding more clarity. Here, we are indicating that because λ values were high, which indicate that*
*physico-climatic variables represented trends in hydrological variables, we might choose from*
*either set of canonical correlations. Although λ2 is slightly lower, the second canonical*
*correlations for the hydrological variables (W2) were higher, and since we were interested in*
*predicting these variables, we opted to use the second canonical correlations for physico-climatic*
*variables (V2) in the regression. The paragraph now reads: "The canonical coefficients from the*
*CCA were acceptably high at λ1 0.97 and λ2 0.77, respectively, indicating that the physico-*
*climatic variables exhibited influence on the hydrological variables (Cavadias et al., 2001; Spence*
*and Saso, 2005). Mean canonical correlation values between the hydrological variables and W2*
*were greater than those with W1 (Table 2); thus, the physico-climatic variables strongly*
*associated to second canonical correlation (i.e., V2) were used in the multiple regressions." (16,*
*452)*
Line 311. What correlation value would indicate "strong"? Does it have a statistical level of significance,
like in the standard correlation analysis? Does a negative value indicate negative correlation? Please
explain.
Comment: The minor modification in the sentences does not specifically address the comment.

*We evaluate the strength of canonical correlations based on Cavadias et al. (2001) as those*
*above 0.75. The specific method did not evaluate a level of significance, as is the case for other*
*correlation analyses. A negative correlation value between the physico-climatic variables and a*
*canonical component (e.g., V2 and Area, Table 2) describes the relationship in the canonical*
*space, and does not necessitate a negative relationship with Q2 or Q100. The influential of the*
*physico-climatic variables on those hydrological variables was determined by the multiple*
*regression.*
Line 311-312. It is true that the correlation value is strong between Q100 (1:100 flow) and W2, but it is
weak for Q2 (mean annual flow) and W2. On the other hand Q2 and W1 has a strong correlation. Also the
lambda value is much greater for V1-W1 combination than for V2-W2 combination. Given that, why was
W2 chosen? Is it because the classification is designed for 1:100 flood prediction? Please provide an
explanation.
Comment: This is not true. For Q2, W1 has a stronger correlation than W2. Please provide an objective
explanation in the texts.
*Please refer to the change mentioned for (Line 310). We chose to use V2 and W2 because both*
*of the hydrological variables exhibited adequate relationships with W2 and a selection of physico-*
*climatic variables were related to V2. In opposition, V1 was not associated with many of the*
*variables, with the highest being 0.64.*
Line 347. What are the "PCs from compositional datasets"? Are these different from PC1-PC6 in the
header of Table 3? Please explain.
Comment: I do not see a new figure, or reference to it. Please address the comment.
*We added reference to the compositional datasets to the heading of 4.1.1 to have consistent*
*language. We also reference Table 1 to refer to the compositional datasets and the number of*
*components used in the clustering analysis (17, 474).*
Line 358. "Weaker", not "less strong".
Comment: This has not been revised.
*We thank the reviewer for identifying that this was not changed in text. We have made the*
*change to "weaker". (17, 485)*
Line 472. Are there 11 study watersheds, as indicated in Line 255? If so, is that a high enough number to
examine all seven classes? Please explain.
Comment: This does not address the comment. Please discuss the limitation of using hydrological data
from only 11 watersheds.
*We recognize that there is a limitation in the current approach for the 11 watersheds to represent*
*the watersheds in the cluster analysis, and that this is an approximation of runoff. We have*
*referenced this limitations in the text: "Ideally, a more detailed estimate of runoff for each*

| 429 | *watershed would be a valuable contribution. In the current study, we used the CCA and eleven* |
| 430 | *reference stations to approximate runoff values for the clustering watersheds. Given the number* |
| 431 | *of watersheds included in the analyses, the diversity of physical characteristics and potential* |
| 432 | *hydrological behaviour is likely not completely represented in the small sample size of available* |
| 433 | *hydrometric stations and represents a limitation of this approach." (28, 829). We also note the* |
| 434 | *limitation and potential impact to uncertainty in our methods, as per request be Referee #1 (12,* |
| 435 | *335).* |
| 436 | |
| 437 | |
| 438 | Line 637. Geography may not be an appropriate term here, because geography encompasses many |
| 439 | things, not just landforms. I would say topography or landform is more appropriate. |
| 440 | |
| 441 | Comment: This has not been done. Instead, geography has been replaced by physio-geography, which is |
| 442 | likely an incorrect spelling of physiography. Please note that physiography is a broad term including the |
| 443 | effects of climate, topography, hydrology, and all other variables in physical geography. |
| 444 | |
| 445 | *We thank the reviewer for their insight on the use of this term. We have revised the use of these* |
| 446 | *terms throughout the manuscript for consistency.* |
| 447 | |
| 448 | |

[revised manuscript text omitted]

---

## Author Response (AR3)

**Response to reviewers: "Watershed classification for the Canadian Prairies"**

*Please note that we have changed the manuscript title to: "A WATERSHED CLASSIFICATION APPROACH THAT LOOKS BEYOND HYDROLOGY: APPLICATION TO A SEMI-ARID, AGRICULTURAL REGION IN CANADA".*

Page and line numbers are shown for each change in reference to the marked-up version.

**Response to Editor**

On P. 10, line 263 of the marked up version, it says "as the name suggests". The name is no longer given, just the symbol, so perhaps this statement should be removed.

> *We have removed this statement as per the Editor's suggestion (10, 263).*

On P. 9, line 246, wetland density seems to be defined as the number of wetlands within the watershed, but should be defined as number of wetlands per unit area (i.e. P. 21, line 611).

> *We thank the Editor for this comment. We have adjusted the sentence to include reference to unit area: "the number of wetlands within the watershed per unit area (i.e., wetland density (km-2))." (9, 246).*

Typo on P. 26, line 751 in the spelling of Baulch.

> *Thank you and we have made the correction.*

In the acknowledgements, you might mention the Global Water Futures Program, for which the CFREF grant was awarded, and since both CFREF and Prairie Water are noted here.

> *We have included the acknowledgement of the Global Water Futures program in this section.*

Note that the map in Fig. 1 includes a north arrow pointing up, but the direction of north varies in the map projections for both the main and inset maps. (The same may apply to Fig. 5.)

> *The projection for both the main and inset maps is the same. However, to improve clarity, we have added a north arrow and scale bar for the inset map. Fig. 5 does not have an inset map, so the map remains unchanged.*

[revised manuscript text omitted]